# The TIR-domain containing effectors BtpA and BtpB from *Brucella abortus* impact NAD metabolism

Julia María Coronas-Serna[1], Arthur Louche[2], María Rodríguez-Escudero[1], Morgane Roussin[2], Paul R. C. Imbert[2], Isabel Rodríguez-Escudero[1], Laurent Terradot[2], María Molina[1], Jean-Pierre Gorvel[3], Víctor J. Cid[1]*, Suzana P. Salcedo[2]*

**1** Departamento de Microbiología y Parasitología, Facultad de Farmacia, Universidad Complutense de Madrid and IRYCIS, Madrid, Spain, **2** Laboratory of Molecular Microbiology and Structural Biochemistry, Centre National de la Recherche Scientifique UMR5086, Université de Lyon, Lyon, France, **3** Aix-Marseille University, CNRS, INSERM, CIML, Marseille, France

☯ These authors contributed equally to this work.
* vicjcid@ucm.es (VJC); suzana.salcedo@ibcp.fr (SPS)

**Data Availability Statement:** All relevant data are within the manuscript and its Supporting Information files.

## Abstract

*Brucella* species are facultative intracellular Gram-negative bacteria relevant to animal and human health. Their ability to establish an intracellular niche and subvert host cell pathways to their advantage depends on the delivery of bacterial effector proteins through a type IV secretion system. *Brucella* Toll/Interleukin-1 Receptor (TIR)-domain-containing proteins BtpA (also known as TcpB) and BtpB are among such effectors. Although divergent in primary sequence, they interfere with Toll-like receptor (TLR) signaling to inhibit the innate immune responses. However, the molecular mechanisms implicated still remain unclear. To gain insight into the functions of BtpA and BtpB, we expressed them in the budding yeast *Saccharomyces cerevisiae* as a eukaryotic cell model. We found that both effectors were cytotoxic and that their respective TIR domains were necessary and sufficient for yeast growth inhibition. Growth arrest was concomitant with actin depolymerization, endocytic block and a general decrease in kinase activity in the cell, suggesting a failure in energetic metabolism. Indeed, levels of ATP and NAD$^+$ were low in yeast cells expressing BtpA and BtpB TIR domains, consistent with the recently described enzymatic activity of some TIR domains as NAD$^+$ hydrolases. In human epithelial cells, both BtpA and BtpB expression reduced intracellular total NAD levels. In infected cells, both BtpA and BtpB contributed to reduction of total NAD, indicating that their NAD$^+$ hydrolase functions are active intracellularly during infection. Overall, combining the yeast model together with mammalian cells and infection studies our results show that BtpA and BtpB modulate energy metabolism in host cells through NAD$^+$ hydrolysis, assigning a novel role for these TIR domain-containing effectors in *Brucella* pathogenesis.

**Funding:** This work was funded by the FINOVI foundation under a Young Researcher Starting Grant, the Cystic Fibrosis French Foundation Vaincre la Mucovicidose grant RF20130500897 and the ANR (grant n°ANR-15-CE15-0011) to SS, and by grants BIO2016-75030-P from Ministerio de Economía y Competitividad (Spain) and S2017/BMD-3691 (InGEMICS-CM) from Comunidad de Madrid and European Structural and Investment Funds to VJC and MM. SS is supported by an INSERM staff scientist contract. J.M.C-S is supported by a predoctoral contract from UCM. The funders had no role in study design, data collection and analysis, decision to publish, or preparation of the manuscript.

**Competing interests:** The authors have declared that no competing interests exist.

## Author summary

*Brucella* is a genus of zoonotic bacteria that cause severe disease in a variety of mammals, ranging from farm animals (as bovines, swine and ovine) to marine mammals. Transmission to humans, often by ingestion of non-treated dairy products, leads to serious systemic infection. *Brucella abortus* invades host cells and replicates intracellularly. Such behavior relies on the injection of bacterial proteins into the host cytoplasm via specialized secretion systems. Our work focuses on the study of two of these factors, BtpA and BtpB, previously described to contain Toll/Interleukin-1 Receptor (TIR)-domains that modulate innate immunity. We use here two biological models: the yeast *Saccharomyces cerevisiae* and human cell lines. We found that the TIR domains of both *Brucella* proteins were necessary and sufficient to collapse energy metabolism in yeast by depleting ATP and NAD+. This result was translatable to higher cells and consistent with the recently described NADase activity of some TIR domains both in mammalian and bacterial proteins. Importantly, we demonstrate that *Brucella* down-regulates total NAD levels in host cells by using both BtpA and BtpB effectors. Our results show that NAD+ is targeted by *Brucella* during infection, which may constitute a novel mechanism for its pathogenicity.

## Introduction

Several bacterial pathogens can circumvent host innate immune responses during infection, often by injecting effector proteins into host cells that target components of innate immune pathways. In many cases, these effectors contain eukaryotic-like domains capable of modulating receptor proximal events. This is the case of Toll/interleukin 1 receptor (TIR) domains present on the cytosolic faces of all Toll-like receptors (TLRs) and corresponding adaptor proteins, enabling the formation of a scaffold for the assembly of intricate protein signaling complexes [1]. The formation of these supramolecular organizing complexes (SMOCs) involves both self-interactions and interactions with other TIR domains [2]. TIR domains are also present in plants, where they mediate disease resistance, in amoebas with a role in ingestion of bacteria and immune-like functions, as well as in many bacterial genera [3].

Several Gram-negative and Gram-positive bacterial pathogens are known to rely on TIR domain-containing protein effectors for down-regulation of TLR-signaling during infection [4]. One of the best characterized is the TIR-containing protein of uropathogenic *E. coli* (TcpC), prevalent in clinical isolates associated with acute pyelonephritis in children. TcpC was shown to contribute to kidney pathology by hijacking the MyD88 TLR adaptor, resulting in inhibition of TLR4 and TLR2 signaling [5]. TcpC inhibition of TRIF- and IL-6/IL-1-dependent pathways has also been described [6]. Interestingly, the observation that expression of TirS from *Staphylococcus aureus*, present in a multi-drug resistant (MDR) island of numerous clinical isolates, is induced by specific antibiotic treatment [7] raises the possibility that these bacterial proteins may be tightly regulated, enhancing virulence, persistence or dissemination in particular clinical contexts such as exposure to selective pressure.

For some pathogens, additional functions have been assigned to bacterial TIR domains other than the downregulation of TLR pathways, as in the case of PumA from *Pseudomonas aeruginosa*, which interferes with TNF receptor signaling by targeting UBAP1 [8], a component of the endosomal-sorting complex required for transport I (ESCRT-I). Also, *E. coli* TcpC directly interacts with the NACHT leucine-rich repeat PYD protein 3 (NLRP3) inflammasome and caspase-1 resulting in inflammasome perturbation [9].

Recent work on mammalian TLR adaptor SARM1 and plant nucleotide-binding leucine-rich repeat (NLR) immune receptors, such as RUN1, unveiled that their TIR domains possess enzymatic activity [10, 11]. Authors went on to demonstrate that not only eukaryotic but also prokaryotic TIR domains, in general, constitute a new family of nicotinamide adenine dinucleotide (NAD⁺) hydrolase enzymes [12]. Although this NADase activity is efficiently neutralized in the bacteria by an unknown mechanism, when heterologously expressed in laboratory *E. coli* strains or assayed *in vitro* these prokaryotic TIR domains were able to cleave NAD⁺. Loss of NAD⁺ was also detected when full-length *S. aureus* TirS was ectopically expressed in mammalian cultured cells [12].

One of the bacterial TIR domains shown to have NAD⁺-consuming activity when expressed in *E. coli* was that of BtpA (also known as TcpB) from *Brucella* spp. [12]. In *Brucella abortus*, a clear role in virulence has been established not only for BtpA but also for BtpB, the second TIR domain-containing protein of *Brucella*. Together, these effectors have been shown to down-modulate dendritic cell activation contributing to the stealthy characteristics of this pathogen in the context of chronic brucellosis [13, 14].

The precise target of *Brucella* TIR-containing effector proteins remains unclear. BtpA has been proposed to act as a mimic of the TLR adaptor TIRAP by binding specific phosphoinositides of the plasma membrane [15] and increasing TIRAP ubiquitination and degradation during infection [16]. However, preferential binding to MyD88 was also demonstrated [17]. It is likely that these *Brucella* TIR-containing proteins display additional targets or functions, as they modulate microtubule dynamics when ectopically expressed [18, 19] and BtpA was shown to induce the unfolded protein response [20].

Given all the possible roles proposed for these *Brucella* TIR effectors and their potential NADase activity we set out to investigate in greater detail their functions. By combining ectopic expression in the model eukaryotic organism *Saccharomyces cerevisiae* and human cells, as well as *in vitro* infection studies we have found that BtpA and BtpB reduce total NAD levels during infection, suggesting their NADase activities are an integral part of their role in *Brucella* pathogenesis. Our results point towards a novel function of these effectors in modulation of host metabolism through the modulation of intracellular NAD levels during infection.

## Results

### Expression of *Brucella abortus* TIR-domain containing BtpA and BtpB proteins in *S. cerevisiae* induces toxicity

To gain insight into the roles of BtpA and BtpB in modulation of cellular functions, *btpA* and *btpB* genes were cloned in a yeast expression vector under the control of the inducible *GAL1* promoter to produce the corresponding GFP fusion proteins. Thus, expression was repressed in glucose-based media, but incubation of yeast transformants in galactose-based media led to the expression of GFP-BtpA and GFP-BtpB, as verified by Western blotting (S1A and S1B Fig). Both GFP-BtpA and GFP-BtpB were inhibitory for yeast growth, but expression of the latter was much more toxic (Fig 1A). At the fluorescence microscope, GFP-BtpB displayed a punctate cytoplasmic pattern, whereas GFP-BtpA was clearly enriched in yeast nuclei (Fig 1B and 1C). In sum, when expressed in yeast, *Brucella* TIR-containing domain proteins lead to different degrees of toxicity and distinct subcellular localization.

### TIR domains of BtpA and BtpB are necessary and sufficient for toxicity, and form filamentous structures in the yeast cell

BtpA and BtpB have divergent N-terminal regions, while their C-terminal fractions display their respective TIR domains. To learn whether cytotoxicity relied on their TIR domains or

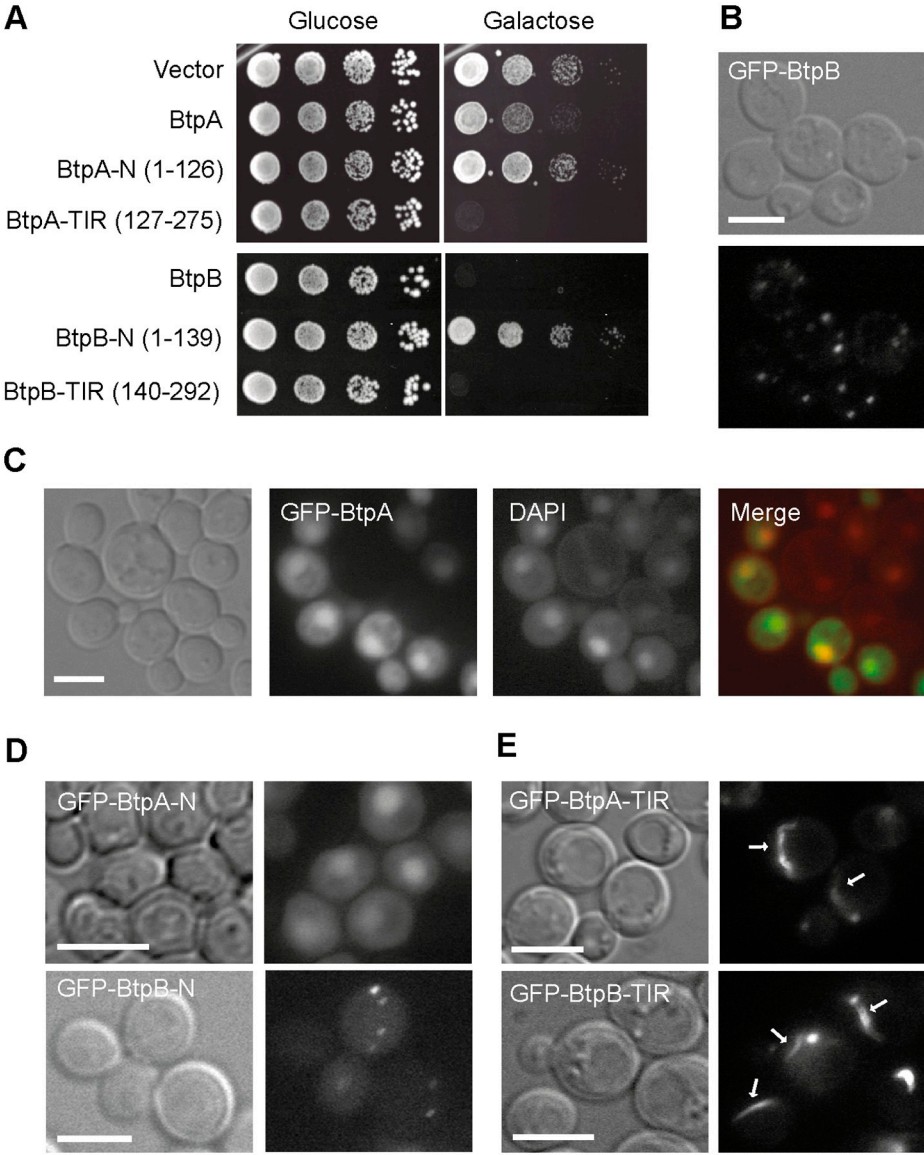

**Fig 1. Expression and localization of *B. abortus* BtpA and BtpB in *S. cerevisiae*.** (A) BtpA and BtpB induce different levels of toxicity when expressed in yeast. Ten-fold serial dilution assay to monitor growth in YPH499 yeast strain expressing the pYES2 empty vector or the BtpA or BtpB indicated versions (full-length, N and TIR) from pYES2-GFP plasmid derivatives, under control (Glucose) and induction (Galactose) conditions. Nomarski and fluorescence microscopy of YPH499 yeast strain expressing from pYES2 plasmid derivatives the following fusion proteins: (B) full-length GFP-BtpB after 4h induction; (C) full-length GFP-BtpA (green) and stained with DAPI (red), after 6h induction; (D) the GFP-fused N-terminal regions of BtpA and BtpB after 5h induction; (E) the GFP-fused C-terminal regions containing TIR domains of BtpA and BtpB after 4h induction. Scale bars correspond to 5 μm.

their N-terminal extensions, we split both proteins to individually express the N- and C-termi-nal halves of the proteins. Thus, we produced GFP fusions to BtpA-N (1–126) and BtpA-TIR (127–275), and BtpB-N (1–139) and BtpB-TIR (140–292) and confirmed their expression in yeast cells (S1A and S1B Fig). As shown in Fig 1A, both TIR domains alone were sufficient for toxicity. Interestingly, the TIR domain of BtpA was more toxic than the full-length protein. In contrast, the N-terminal regions of BtpA and BtpB were innocuous for the yeast cell. However,

the N-terminal extensions defined subcellular localization of the proteins, as BtpA-N maintained the predominant nuclear localization and BtpB-N formed cytoplasmic dots, like the corresponding full-length proteins (Fig 1D). In spite of the limited identity (19.92%) between the TIR domains of BtpA and BtpB, fluorescence microscopy revealed that both GFP-BtpA-TIR and GFP-BtpB-TIR assembled into long cytoplasmic filaments (Fig 1E), occasionally contacting or surrounding the nucleus (S2 Fig). Both GFP-BtpA-TIR and GFP-BtpB-TIR conspicuous filaments resembled cytoplasmic microtubule bundles. However, immunofluorescence with anti-tubulin antibodies revealed that they did not co-localize with tubulin (S2 Fig). Although we cannot rule out that *Brucella* TIR domains are interacting with filamentous structures other than tubulin in yeast, our results suggest that they are prone to forming highly ordered structures by self-interaction, and that their N-terminal extensions negatively influence this behaviour.

## BtpB depolarizes actin patches, blocks endocytosis and down-regulates signaling in *S. cerevisiae*

To understand the mechanisms underlying growth inhibition in yeast expressing *Brucella* TIR proteins, we chose to analyze the effects of full-length BtpB. The actin cytoskeleton supports polarized growth in yeast during budding, so growth arrest could be caused by actin dysfunction. Indeed, as shown in Fig 2A, staining of actin cortical patches with rhodamine-conjugated phalloidin revealed a dramatic loss of polarization of actin structures towards the growing bud and septum region. Moreover, the BtpB TIR domain was fully responsible for this phenotype (Fig 2A). Besides supporting growth along the budding cycle, actin function is important for endocytosis. We used the FM4-64 fluorochrome to monitor endocytic traffic. Internalization of this non-permeable molecule via the endocytic pathway leads to staining of the vacuolar membrane in cells after 1 hour of incubation [21]. We observed that cells that efficiently expressed GFP-BtpB, as judged by the presence of intense green fluorescent cytoplasmic spots, were unable to internalize this marker, as compared to those lacking green fluorescence or control cells expressing GFP alone (Fig 2B), indicating that BtpB severely blocked endocytosis. This phenotype also relies on BtpB TIR domain, as shown in S1C Fig.

Often, cellular stresses that lead to actin depolarization in yeast trigger the activation of signaling cascades involving mitogen-activated protein kinase (MAPK) modules, such as the cell wall integrity (CWI) pathway, engaging the Slt2 MAPK [22]. Also, we have previously described that some bacterial effectors, such as *Salmonella* SteC and SopB [23, 24], depolarize actin by down-regulating small GTPases when expressed in yeast, leading to concomitant dephosphorylation of downstream Fus3 and Kss1 MAPKs of the mating pathway. Thus, we tested MAPK activation levels in BtpB-expressing cells by immunoblot using anti-phospho-MAPK antibodies. Peculiarly, all Slt2, Fus3 and Kss1 MAPK basal phosphorylation levels were downregulated in BtpB-expressing cells, but not upon BtpA expression (S3A Fig). Then we investigated whether BtpB would be able to downregulate MAPK activation upon stimulation of these pathways, by incubation at 39 ºC or in the presence of the cell wall-stressing compound Congo red to stimulate the CWI pathway, and by using the mating pheromone α-factor to activate Fus3 and Kss1. Although BtpB still allowed activation of these pathways by the stimuli, MAPK phosphorylation was always less efficient (S3B Fig). A fourth MAPK, Hog1, a p38 homolog, operates in budding yeast responding to high osmolarity challenges [25]. As observed for the other MAPKs, phosphorylation of Hog1 was less efficient in the presence of BtpB when this pathway was stimulated by osmotic stress (S3C Fig). Since these MAPK pathways do not share upstream components, it is striking that they all were simultaneously downregulated by BtpB expression. Such a general effect in MAPK phosphorylation might reflect

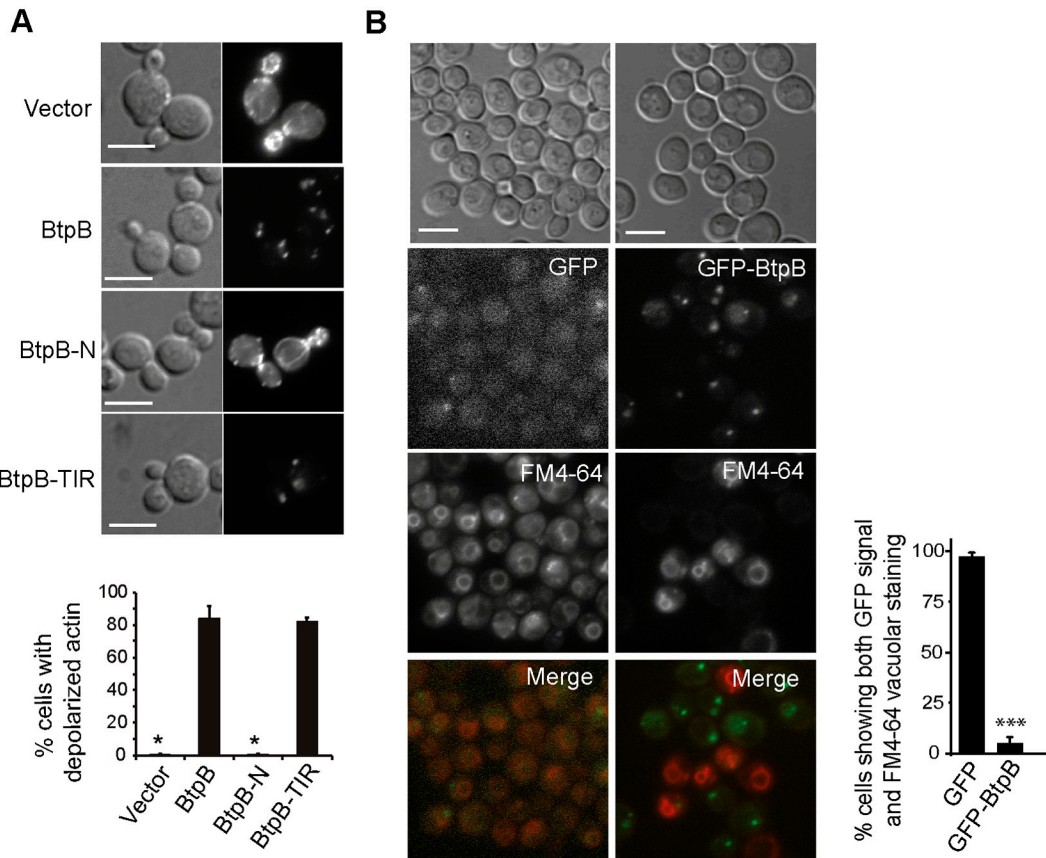

**Fig 2. BtpB expression causes severe defects in actin cytoskeleton and endocytosis in *S. cerevisiae*.** (A) Nomarski and fluorescence microscopy images (upper panel) and graph showing the percentage of small- to medium-budded cells with depolarized actin (lower panel) after rhodamine-phalloidin staining of YPH499 cells expressing pYES2-GFP empty vector, BtpB, BtpB-N or BtpB-TIR from pYES2-GFP plasmid derivatives after 4h induction. Data correspond to means ± standard deviation of three independent transformants (n ≥ 100) and statistical comparison was done with Kruskall-Wallis ANOVA with p-values referring to BtpB of 0.024 (*) for vector and BtpB-N. Scale bars indicate 5 μm. (B) Nomarski and fluorescence microscopy images (left panel) and graph representing the percentage of cells showing both GFP and FM4-64 vacuolar signal (right panel) of YPH499 cells expressing pYES2-GFP or pYES2-GFP-BtpB, after 4h induction, stained with the endocytic marker FM4-64 for 1h. Data correspond to means ± standard deviation of three independent transformants (n ≥ 100) and statistical comparison was done with Student´s t-test, p < 0.0001 (***). Scale bars indicate 5 μm.

inability of the cell to properly undergo phosphorylation events. In support of this view, when heterologous mammalian Akt1 (which undergoes phosphorylation in its activation site by conserved yeast PDK-like kinases [26]), was co-expressed with BtpB, a reduced phosphorylation was also observed (S3D Fig). The TIR domain of BtpB alone was fully responsible for signaling down-regulation (S1B Fig). Interestingly, such effect was also observed when expressing the TIR domain of BtpA (S1A Fig).

In sum, BtpB and BtpB-TIR expression in yeast result in severe actin disorganization, endocytic block and a general defect in the phosphorylation of all signaling kinases tested.

## Genetic screen for yeast genes that suppress BtpB-induced lethality

We pooled three non-overlapping libraries obtained from the whole genome yeast ORF collection, consisting of all *S. cerevisiae* predicted ORFs cloned in an expression vector under the

control of the inducible *GAL1* promoter transformed in *E. coli*. This pooled whole genome expression library was co-transformed with a GFP-BtpB *GAL1*-based expression plasmid and positive selection allowed the recovery of genes suppressing BtpB toxicity in galactose-based medium. Suppressor genes listed in S4A Fig and S3 Table were selected when growth rescue (i) was confirmed after individual re-transformation, (ii) was specific for BtpB-induced growth inhibition, but not that of other toxic heterologous protein (PI3Kα-CAAX)[26], and (iii) was not due to a lower production of GFP-BtpB, as verified by immunoblot (S4B Fig). As shown in S4A Fig, suppression was partial in all cases. Co-transformation of these suppressors with BtpB-TIR led to the same rescue levels, although no growth recovery was detected when co-expressed with BtpA-TIR (S4C and S4D Fig). Thus, either these suppressors are specific for BtpB-TIR domain derived toxicity in yeast or the effect of BtpA-TIR is too strong to allow partial suppression. Although most of these genes have not been yet assigned a bona fide function in yeast, a subset of them, *INM2*, *RBK1*, and *DOG2* are sugar or inositol phosphorylating/dephosphorylating enzymes related to metabolic pathways (S3 Table). *DOG2* encodes a 2-deoxyglucose-6 phosphate phosphatase and its overexpression overcomes toxicity of this glycolytic inhibitor [27], and *RBK1* encodes a putative ribokinase, which has been recently shown to be catalytically active [28]. These results suggest that metabolic shifts related to carbon source usage partially counteract BtpB toxicity.

## BtpA and BtpB deplete ATP and NAD+ in the yeast cell

Our results on BtpB expression in yeast affecting dynamic cellular events such as cytoskeletal function and vesicle traffic as well as general kinase function would be consistent with limiting intracellular ATP levels. Furthermore, the fact that sugar kinase/phosphatases were isolated as *btpB* overexpression suppressors suggest that energetic metabolism is compromised in BtpB-expressing yeast cells. Recently, Essuman *et al.* [12] reported that the TIR-domain of proteins from phylogenetically diverse bacteria, including *Brucella* BtpA, displayed enzymatic activity as NAD+ hydrolases. Thus, we were prompted to study ATP and NAD+ levels in yeast cells expressing BtpA and BtpB. Yeast cells expressing BtpB or the TIR domains of either BtpB or BtpA, showed significant losses of both ATP and NAD+ intracellular levels, as determined by luciferase assay or quantitative mass spectrometry respectively (Fig 3). This effect was especially dramatic in NAD+ levels, which were lowered about one order of magnitude upon BtpB overexpression. We also observed a slight but significant reduction of NAD+ in the case of full-length BtpA (Fig 3B). The decrease in NAD+ and ATP correlated very well with the differential toxicity for yeast cells of each protein version (Fig 1A), as full-length BtpB had the strongest effect on intracellular NAD+ and ATP levels while, in the case of BtpA, the TIR domain alone had a more dramatic effect than the full-length protein. This raises the idea that the N-terminal extension of BtpA, but not that of BtpB, has a negative regulatory effect on the C-terminal TIR/NAD+ hydrolase domain.

As a control, we generated a BtpB E234A catalytically inactive mutant, by changing to Ala the equivalent Glu residue described by Essuman *et al.* [12] to be essential for catalysis in other TIR domains. BtpB E234A was no longer toxic for yeast (Fig 4B), and it did not lead to reduced MAPK phosphorylation (S1B Fig), or endocytosis defects (S5D–S5E Fig). In agreement with its lower toxicity, this BtpB mutant was expressed at higher levels than the toxic wild-type version (S1B Fig). Expression of the BtpB E234A mutant had no effect on ATP or NAD+ intracellular levels (Fig 3), strongly suggesting that, as described for other TIR domains [12], this residue is essential for the catalytic activity of BtpB.

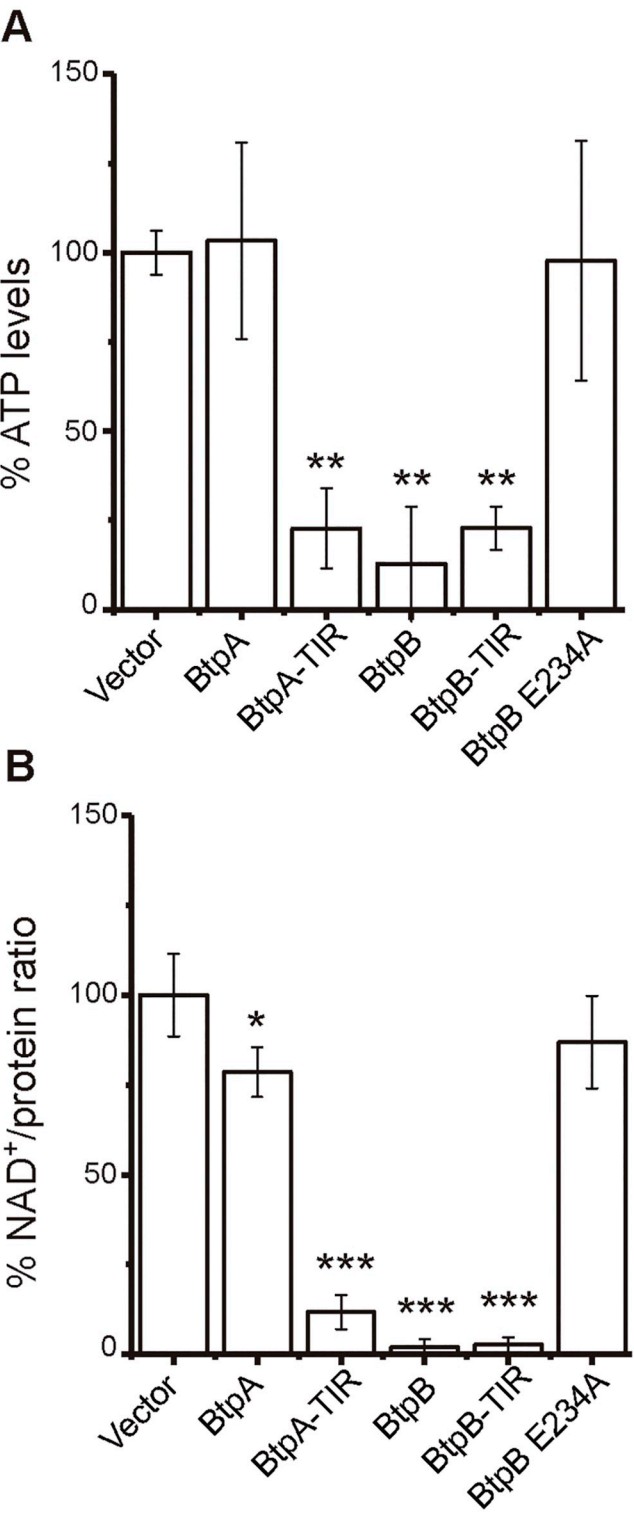

**Fig 3. BtpB, BtpA-TIR and BtpB-TIR reduce NAD⁺ and ATP levels when expressed in yeast.** (A) Cellular ATP measurement by luciferase assay in YPH499 cells transformed with pYES2 empty vector and pYES2 plasmid derivatives bearing: both full-length and TIR domain versions of BtpA and BtpB and the catalytically inactive BtpB E234A mutant. Graph shows ATP levels as a percentage relative to the ATP levels measured on empty vector control cells. Results correspond to means ± standard deviation of three different transformants and statistical comparison was done with one-way ANOVA with p-values referred to vector of 0.0045 (**) for BtpA-TIR, 0.0017 (**) for BtpB and

0.0046 (**) for BtpB-TIR. (B) Cellular NAD$^+$ levels measured by mass spectrometry, standardized as a NAD$^+$/extract protein ratio, from YPH499 cells transformed with the same plasmids as in A. Graph shows NAD$^+$/protein ratio as a percentage of the empty vector control cells NAD$^+$/protein ratio. Data correspond to means ± standard deviation of four different transformants and statistical comparison was done with one-way ANOVA with p-values referred to vector <0.0001 (***) for BtpA-TIR, BtpB and BtpB-TIR, and 0.0134 (*) for BtpA.

## Mapping of residues essential for NAD$^+$ hydrolase function at the TIR domain of BtpB

Taking advantage of the severe toxicity of BtpB in yeast, we devised a screen for the isolation of loss-of-function mutations by random mutagenesis. This was performed by plasmid gap-repair [29], forcing *in vivo* homologous recombination between an open gapped plasmid and a partially overlapping insert encoding BtpB, which had been generated by error-prone PCR. Such strategy allows direct selection in galactose-based medium for recombinant clones bearing mutations in *btpB* that yield the protein non-toxic. Ten single and two double mutants were recovered and sequenced (S4 Table). As shown in S5A Fig, most mutations corresponded to non-conservative amino acid changes in highly conserved regions between BtpA and BtpB TIR domains. Some of these residues are also conserved in the TIR-domain of human SARM1 and plant RUN1, in which the NAD$^+$ hydrolytic activity was recently described [10, 11].

To decipher the effects of the mutations on BtpB properties, we mapped the corresponding residues on the BtpA-TIR domain structure (PDB: 4LZP) [30], as BtpB structure is not yet solved. As seen in Fig 4A, none of the residues mutated belonged to the TIR-TIR interface. S162 (S149 in BtpA) and F163 (H150 in BtpA) belong to the βA strand. Y225 (F208 in BtpA) and Q226 (F209 in BtpA) are part of the small helix αC. Mutations of these residues are likely to disrupt the inner core and thus destabilize the whole structure.

Mutation of BtpB S201P (S185 in BtpA) is likely to perturb the NAD+ catalytic site. In the recent crystal structure of NADP$^+$-bound RUN1-Tir domain [11] (PDB: 6O0W), the substrate lies in a pocket formed by the BB-loop and the loop containing the conserved catalytic WxxxE motif [19] (S5B Fig). In BtpA structure S185 lies in the BB loop and interacts with the W213 (W231 in BtpB) of the WxxxE motif, which contains the essential catalytic E217 residue (E234 in BtpB). Finally, D158 (D145 in BtpA), F188 (F174 in BtpA), Y193 (Y178 in BtpA), and I291 (I275 in BtpA) residues clustered in two patches at the protein surface (Fig 4A).

In order to determine whether NAD+ hydrolase and filament formation of TIR domains were separable features, we transferred mutations D158G, S162P and Y255C, as well as the mutation in the catalytic residue E234A, to GFP-BtpB-TIR to study whether loss of toxicity correlated with the ability of the TIR domain alone to produce filaments. Interestingly, only E234A, S162P and, partially, Y225C mutations eliminated BtpB-TIR toxicity in yeast cells (Fig 4B), despite the fact that all four mutations fully prevented toxicity and endocytosis defects in full-length BtpB (S5C–S5E Fig). Moreover, only the GFP-BtpB-TIR S162P mutant significantly lost the ability to form protein filaments (Fig 4C and 4D), probably because that mutation damaged the inner core (Fig 4A) These results indicate segregation between filament formation and growth inhibitory functions of BtpB-TIR in yeast and highlight the importance of the Glu234 residue specifically for NAD$^+$ hydrolase activity, while Ser162 is key for both features.

We also changed by site-directed mutagenesis the catalytic E217 residue to Ala in both full-length and the TIR domain alone of BtpA. Both mutants lost their toxicity on yeast (Fig 4E). Moreover, BtpA-TIR E217A did not reduce MAPK phosphorylation and yeast cells sustained higher levels of expression as compared to WT BtpA-TIR (S1A Fig). Although a statistically significant reduction in the percentage of cells showing BtpA-TIR filaments was found (45.1%

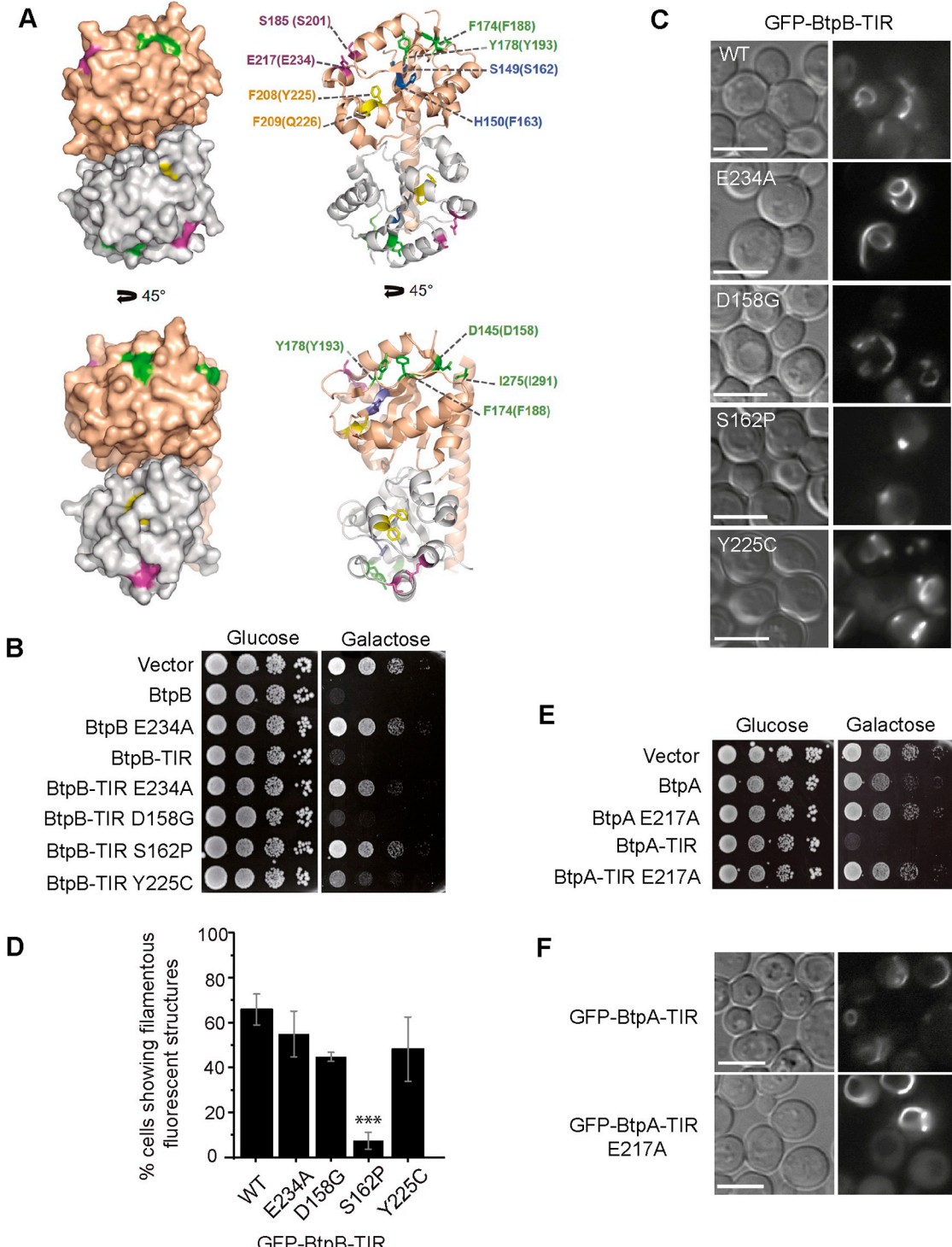

**Fig 4. Functional analysis of BtpB and BtpA mutations in yeast.** (A) Structure of BtpA-TIR domain showing the positions equivalent to those identified as loss-of-function in BtpB by yeast random mutagenesis screening. Left panels: two views of BtpA-TIR dimer structure (PDB: 4LZP) with chain A colored in wheat and chain B in grey. Residues identified are colored according to their assigned properties. Positions of the mutations putatively affecting protein folding are colored in blue (βA strand) and yellow (αC helix). Mutations at the active site are colored in magenta. Mutations at the surface outside the active site are colored in green. Right panels: views in the same orientation of the BtpA dimer depicted as cartoon with the side chains of mutated residues displayed as ball-and-sticks. Residue numbers are indicated for BtpA and the corresponding residues in BtpB are in parenthesis. (B) Ten-fold serial dilution growth assay of YPH499 cells expressing pYES2 empty vector, BtpB full-length, BtpB-TIR and the indicated mutants

from pYES2 plasmid derivatives, under control (Glucose) and induction (Galactose) conditions. (C) Normarski and fluorescence microscopy of YPH499 cells expressing GFP-BtpB-TIR and the indicated mutants, after 4h induction. Scale bars correspond to 5 μm. (D) Graph displaying percentage of cells showing filamentous fluorescent structures. Data corresponds to means ± standard deviation of three independent transformants and statistical comparison was done with one-way ANOVA with a p-value < 0.0001 (***) between BtpB-TIR WT and S162P. (E) Ten-fold serial dilution growth assay of YPH499 yeast strain bearing pYES2 empty vector and pYES2 plasmid derivatives expressing BtpA, BtpA-TIR and their corresponding catalytically inactive mutants E217A, under control (Glucose) and induction (Galactose) conditions. (F) Normarski and fluorescence microscopy of yeast cells expressing pYES2-GFP-BtpA-TIR and its E217A mutant version. Scale bars correspond to 5 μm.

± 7.7 for BtpA-TIR vs. 26% ± 12 for BtpA-TIR E217A), these structures were larger and more intense for the mutant than for the wild type version (Fig 4F). Importantly, these results indicate that, as observed for BtpB, the catalytic E217 residue is essential for toxicity in yeast but still allows assembly of the BtpA TIR domain into ordered structures.

## Inhibition of endocytosis occurs upon ectopic expression of BtpB in human cells but not during infection

To investigate whether the results obtained in yeast were translatable to mammalian cells, we began by overexpressing BtpB in human epithelial cells (HeLa). As previously described [19] and consistent with the yeast model, a punctate accumulation of BtpB was observed in the cytosol of HeLa cells. These results were obtained for GFP-BtpB (Fig 5), as well as Myc-expressing BtpB (S6A Fig), indicating that this localization is independent of the tag. To gain insight into the type of structures BtpB was forming, we labelled for different endocytic markers. Some of these structures were enriched in mono- and poly-ubiquitinated proteins as recognized by the FK2 antibody (Fig 5A), which could correspond to either aggregates of misfolded protein or sites with densely ubiquitinated proteins as previously described [19]. However, some of the BtpB compartments did not show labelling with the FK2 antibody and were also negative for the lysosomal associated membrane protein 1 (LAMP1) (Fig 5A), suggesting BtpB associates with multiple intracellular structures. Unlike the yeast model, expression of BtpB neither resulted in significant perturbation of the actin cytoskeleton (Fig 5B) nor the microtubule network morphology (Fig 5C). Consistent with results from Felix and colleagues [19], we also observed localization of BtpB between cells, at sites of intercellular bridges that form during cell division (S6B Fig).

As the yeast model revealed a potential role for the N-terminal domain of BtpB in intracellular localization whereas the TIR domain for its toxicity, we next analyzed the fate of truncated BtpB versions in HeLa cells. As in yeast, expression of BtpB-N resembled that of full-length BtpB, with cytosolic aggregates being formed (Fig 6). Expression of the TIR domain alone (BtpB-TIR) resulted in the formation of long filament-like structures that showed no co-localization with tubulin (Fig 6), consistent with the results obtained in the yeast model. In some cells, BtpB-TIR induced disorganization of the microtubule network (S6C Fig). These filamentous structures did not co-localize with vimentin either, a marker of intermediate filaments (S6D Fig), strongly reminiscent of what has been previously described for the *Staphyloccocus aureus* TirS protein [7]. Most likely, as also inferred from the above yeast data, these structures correspond to self-assembled ordered filaments consisting of the TIR domain, which are absent when stabilized by the presence of the N-terminal domain. These results confirm that the N-terminal portion of BtpB plays an important role in subcellular localization. Interestingly, the BtpB E234A mutant retained the dot-like distribution observed for BtpB in HeLa cells (S6E Fig).

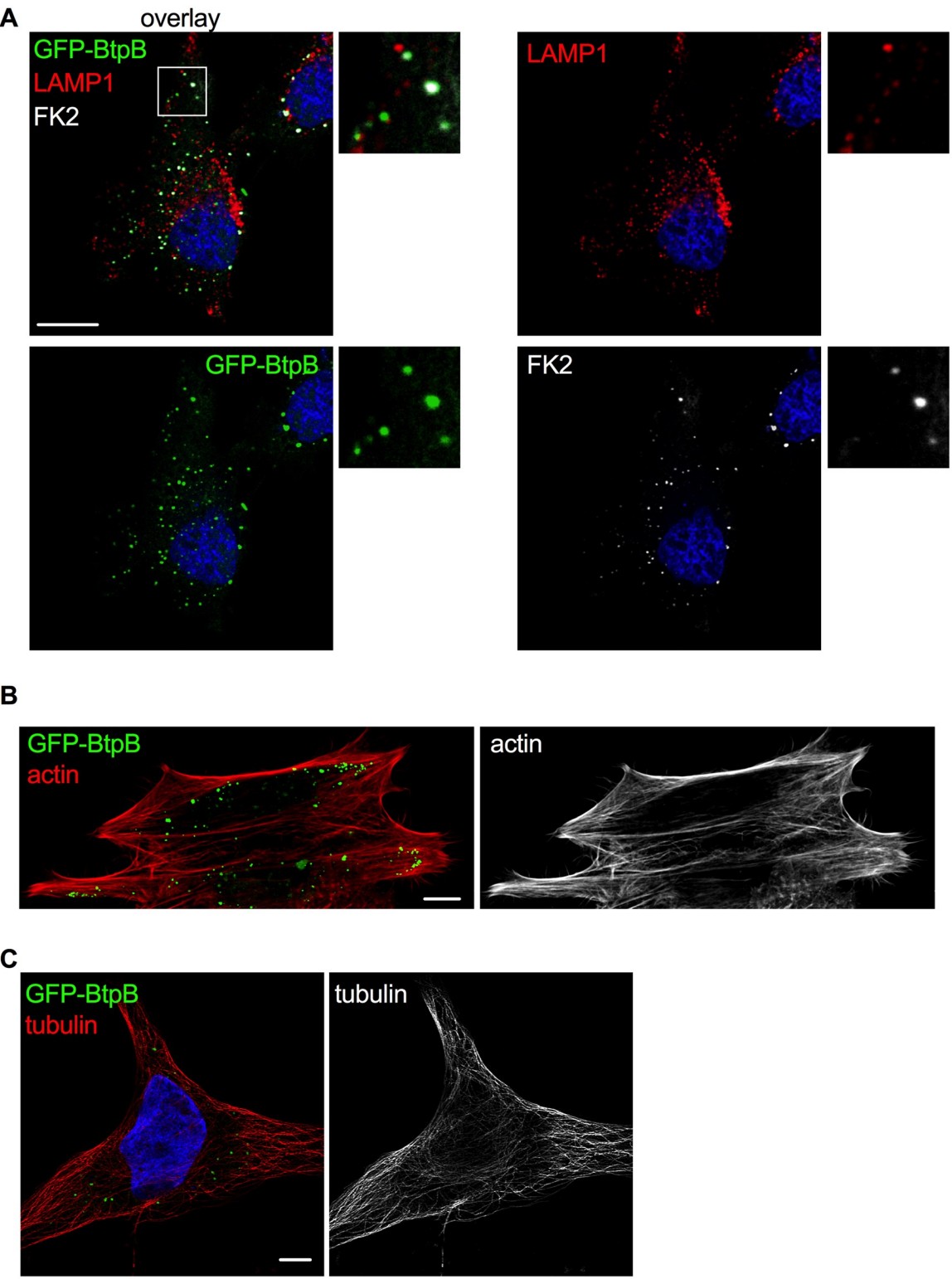

**Fig 5. Localization of ectopically expressed BtpB in human cells.** HeLa cells expressing GFP-BtpB (green) were labelled and analyzed by confocal immunofluorescence microscopy. Representative images are shown. (A) Cells were labelled with FK2 (cyan) along with LAMP1 (red) antibodies. The nuclei are labelled with DAPI (blue). (B) Cells were labelled with phalloidin for visualization of the actin cytoskeleton (red) and (C) with anti-tubulin antibody for visualization of the microtubules (red). Scale bars correspond to 5 μm. The tubulin image was obtained with Airyscan confocal imaging mode.

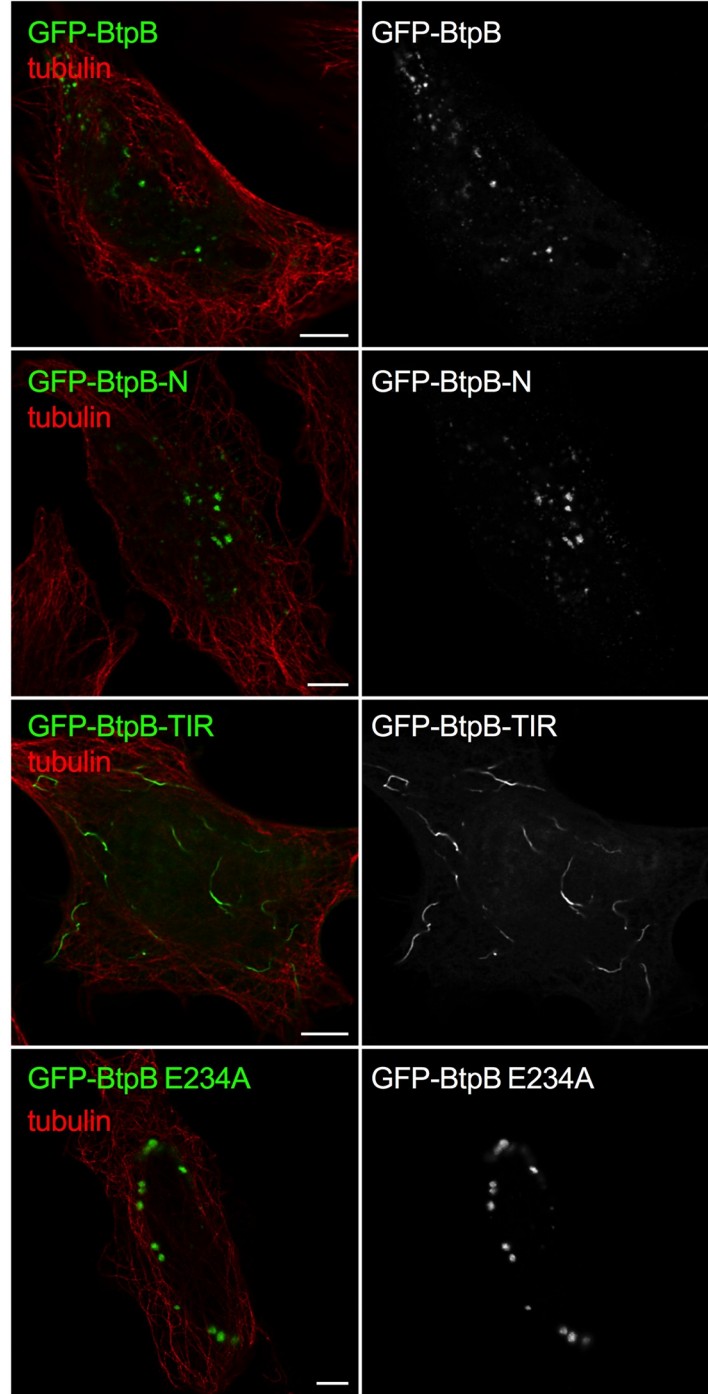

**Fig 6. BtpB N-terminal domain is required for intracellular localization when ectopically expressed. The** HeLa cells were transfected with GFP-BtpB, GFP-BtpB-N (1–139), GFP-BtpB-TIR (140–292) and GFP-BtpB E234A. Cells were then labelled for tubulin (red). Scale bars correspond to 5 μm and all images obtained with Airyscan confocal imaging mode.

We next determined if BtpB expression resulted in perturbation of endocytosis in human cells as observed in yeast. Fluorescently labeled transferrin or Epidermal Growth Factor (EGF) were incubated with HeLa cells expressing GFP-BtpB or GFP alone and the percentage of cells with uptake of these endocytosis markers quantified by microscopy. The expression of GFP-BtpB significantly decreased endocytosis of both markers in comparison to GFP alone (Fig 7A–7C) consistent with the yeast model.

Although overexpression of individual effectors provides a powerful tool to investigate direct functions of these proteins, to thoroughly investigate the capacity of BtpA and BtpB to inhibit endocytosis, we analyzed this phenotype during infection. Although we could observe a slight decrease of endocytosis after 24 h of infection of HeLa cells, this phenotype was neither statistically significant nor abrogated by deletion of *btpA* nor *btpB* (S7A Fig). Furthermore, no significant differences were observed at 48 h post-infection for HeLa cells infected with WT *Brucella* in comparison to cells infected with mutants lacking either *btpA*, *btpB* or both genes (Fig 7D). Finally, no impact on endocytosis was observed in immortalized bone marrow-derived macrophages (iBMDM) at 24 nor 48 h post-infection (S7B Fig and Fig 7E, respectively), suggesting that *Brucella* TIR proteins do not interfere with endocytosis during infection.

## BtpA and BtpB deplete cellular NAD when ectopically expressed in human cells and during infection

As previous studies attributed a NAD⁺-consuming activity to the BtpA TIR domain when expressed in *E. coli* [12] and our experiments using the eukaryotic yeast model showed that both BtpB and BtpB-TIR strongly reduce intracellular NAD⁺ content, we next assessed total NAD levels in HeLa cells expressing either Myc-BtpB or Myc-BtpA in comparison to Myc alone by using a colorimetric assay. Both Myc-tagged proteins were well expressed in epithelial cells although BtpA always migrated as a double band, potentially indicative of post-translational modifications occurring in the cell (Fig 8A). As shown in Fig 8B, both BtpA and BtpB strongly reduced total NAD levels in HeLa cells validating the results obtained with the yeast model.

To determine whether *Brucella* could impact intracellular NAD levels during infection, we first established that all bacterial strains had equivalent levels of total NAD in the inocula (S7C Fig), which corresponds to a 16h culture, the time required to reach early stationary phase used for our infection studies. We next infected HeLa cells with wild-type or mutant strains lacking either *btpA* or *btpB* and quantified the levels of total NAD. Although we did not observe any differences at 24h post-infection, we could observe that *B. abortus* infection for 48h resulted in reduction of total NAD levels in a manner dependent on BtpA and BtpB (Fig 9A). As it is well established that *btp* mutants replicate to the same levels as wild-type *Brucella* [14] and we have found they have equivalent bacterial total NAD levels (S7C Fig) we can conclude that BtpA and BtpB NAD-consuming activities are likely to impact host intracellular NAD levels. To confirm these phenotypes were specifically due to the absence of BtpA and BtpB, we attempted to complement the mutant strains. The phenotype of the *btpA* mutant strain could be restored by expressing *btpA* from a plasmid (Fig 9B). Although the same tendency could be observed for the complementation of the *btpB* mutant, due to a lower effect on the NAD concentration in HeLa cells infected with the *btpB* mutant we could not obtain statistical significance with the number of experiments performed (Fig 9C). We therefore infected iBMDM, as a much higher rate of infection can be attained with phagocytic cells. In this cellular model, wild-type *B. abortus* infection also resulted in the reduction of intracellular NAD levels, in a manner dependent on BtpA and BtpB. The expression of each gene from a plasmid

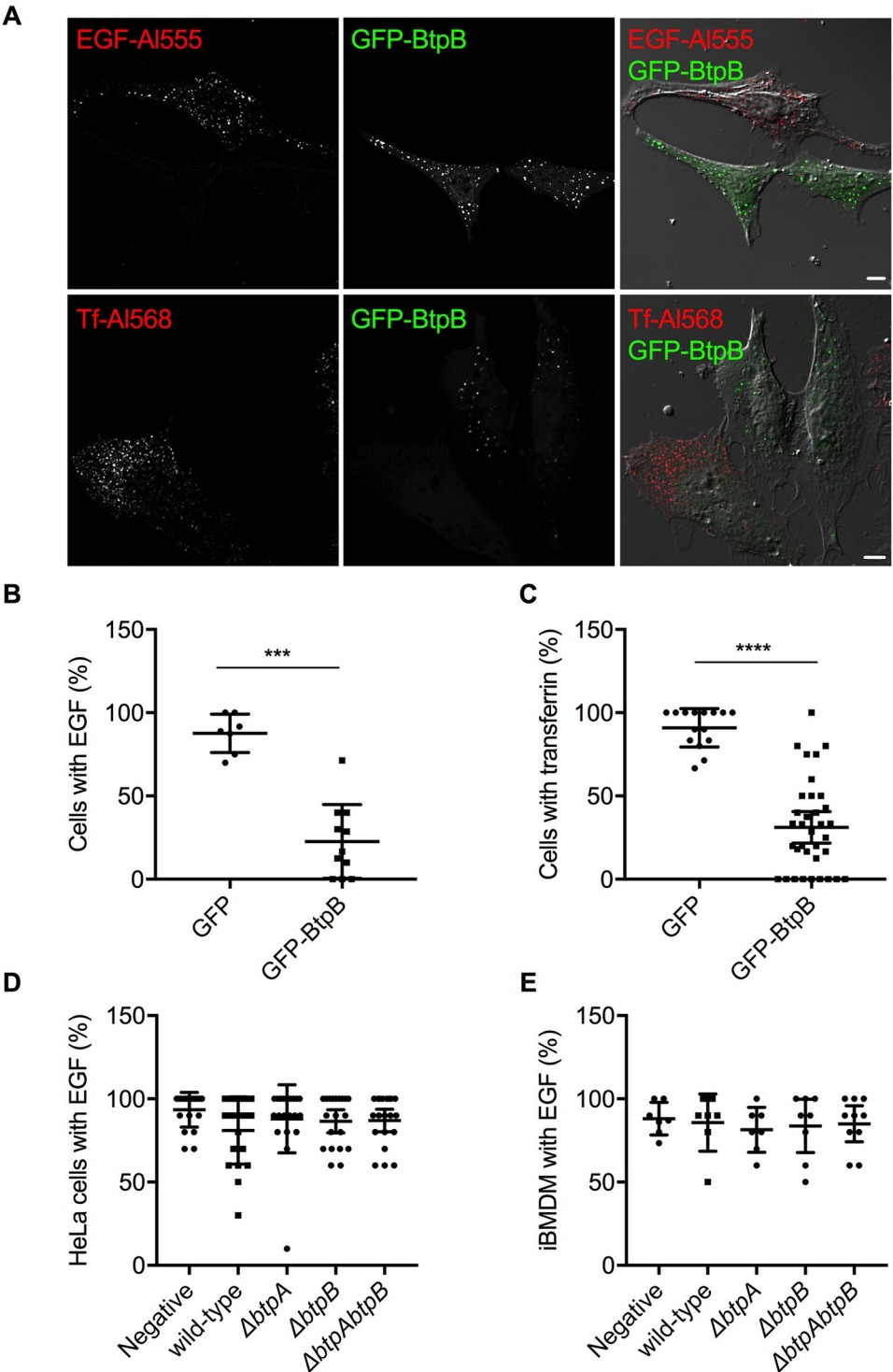

**Fig 7. Inhibition of endocytosis occurs upon ectopic expression of BtpB but not during infection.** HeLa cells expressing GFP-BtpB (green) were incubated with either EGF conjugated with Alexa Fluor 555 (red) or transferrin conjugated with Alexa Fluor 568 (red) for 10 minutes. (A) Cells were then analyzed by confocal microscopy and (B) and (C) the percentage of cells showing uptake of either fluorescent marker quantified. A cell was considered positive when clear labelling of endocytic vesicles was observed throughout the cell. Counts correspond to individual microscopy fields, obtained from three independent experiments. Data correspond to means ± standard deviation and statistical comparison was done with Mann-Whitney test, p = 0.0001 (\*\*\*) for EGF (left) and p < 0.0001 (\*\*\*\*) for transferrin (right). (D) HeLa cells or (E) immortalized bone marrow-derived macrophages (iBMDM) were infected for

48 h with either wild-type *B. abortus* or a mutant strain lacking *btpA*, *btpB* or both genes. Cells were then incubated with EGF conjugated with Alexa Fluor 555 for 10 minutes and the percentage of infected cells showing uptake of this fluorescent marker quantified by microscopy. Counts correspond to individual microscopy fields, with a total of at least 200 cells counted for each, from three independent experiments. Mock infected cells are included as a control. Data correspond to means ± standard deviation and statistical comparison was done with one-way ANOVA test, with no statistical significance observed.

fully restored the wild-type phenotype in the case of BtpA (Fig 10A) and partially in the case of BtpB (Fig 10B).

Finally, to determine if the observed reduction of NAD was due to the catalytic activity of the TIR domain we complemented the *btpA* mutant with a plasmid carrying a E217A mutation in *btpA* (Δ*btpA*p*btpA* E217A) and the *btpB* mutant with a plasmid expressing a E234A mutation in *btpB* (Δ*btpB*p*btpB* E234A). We first controlled that these catalytic mutant versions of BtpA and BtpB could be efficiently translocated into host cells. We constructed TEM1 fusions as previously reported [14] and determined the percentage of cells emitting coumarin fluorescence at 24 h post-infection (S7D Fig). We had to use RAW macrophages for these experiments as previously described [14] because CCF2 was toxic for iBMDM. In the case of BtpA we observed a level of translocation of TEM-BtpA E217A consistent with what was previously reported for the wild-type TEM-BtpA [14]. In the case of BtpB, a lower percentage of infected cells showed translocation of the TEM-BtpB E234A, consistent with what has been observed for the wild-type TEM-BtpB (less than 2% of infected cells) [14].

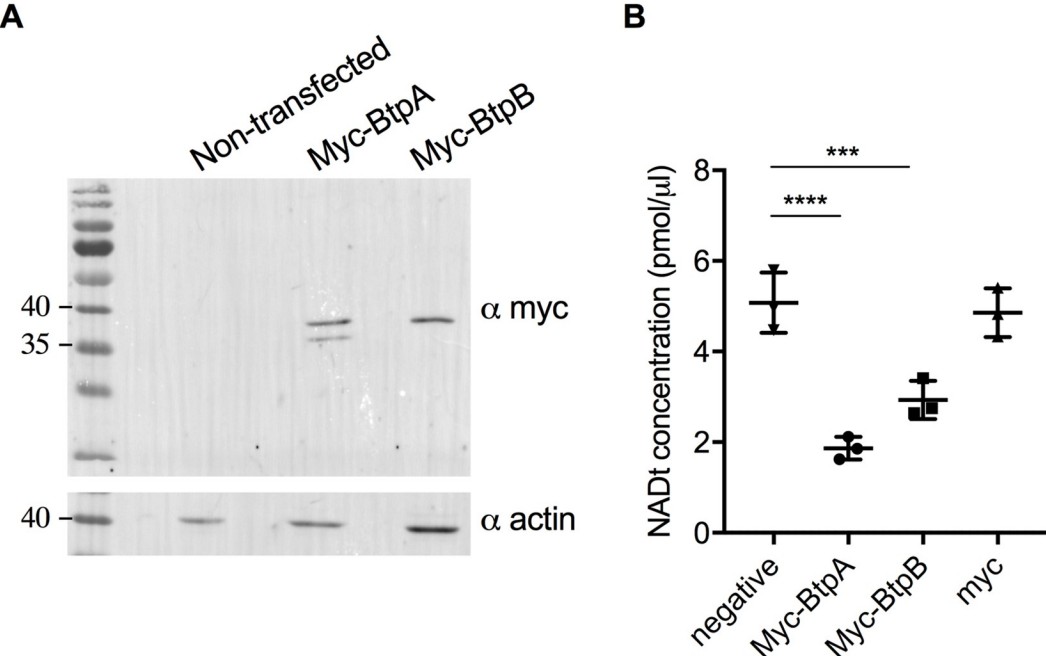

**Fig 8. Ectopic expression of BtpB results in reduction of total NAD⁺ in HeLa cells.** (A) Representative Western blot showing levels of Myc-BtpA and Myc-BtpB expression revealed with an anti-Myc antibody and anti-actin antibodies, as a loading control. Myc-BtpA has a predicted molecular weight of 33 kDa whereas BtpB 38 kDa. (B) Measurement of total NAD levels using a colorimetric assay from HeLa cells expressing either Myc-tagged BtpA or BtpB. Non-transfected cells (negative) and cells transfected with Myc vector alone are also included as controls. Data correspond to means ± standard deviation from three independent experiments and statistical comparison was done with one-way ANOVA, with a p-value for the negative control *versus* Myc-BtpA of <0.0001 (****) and *versus* Myc-BtpB of 0.0006 (***).

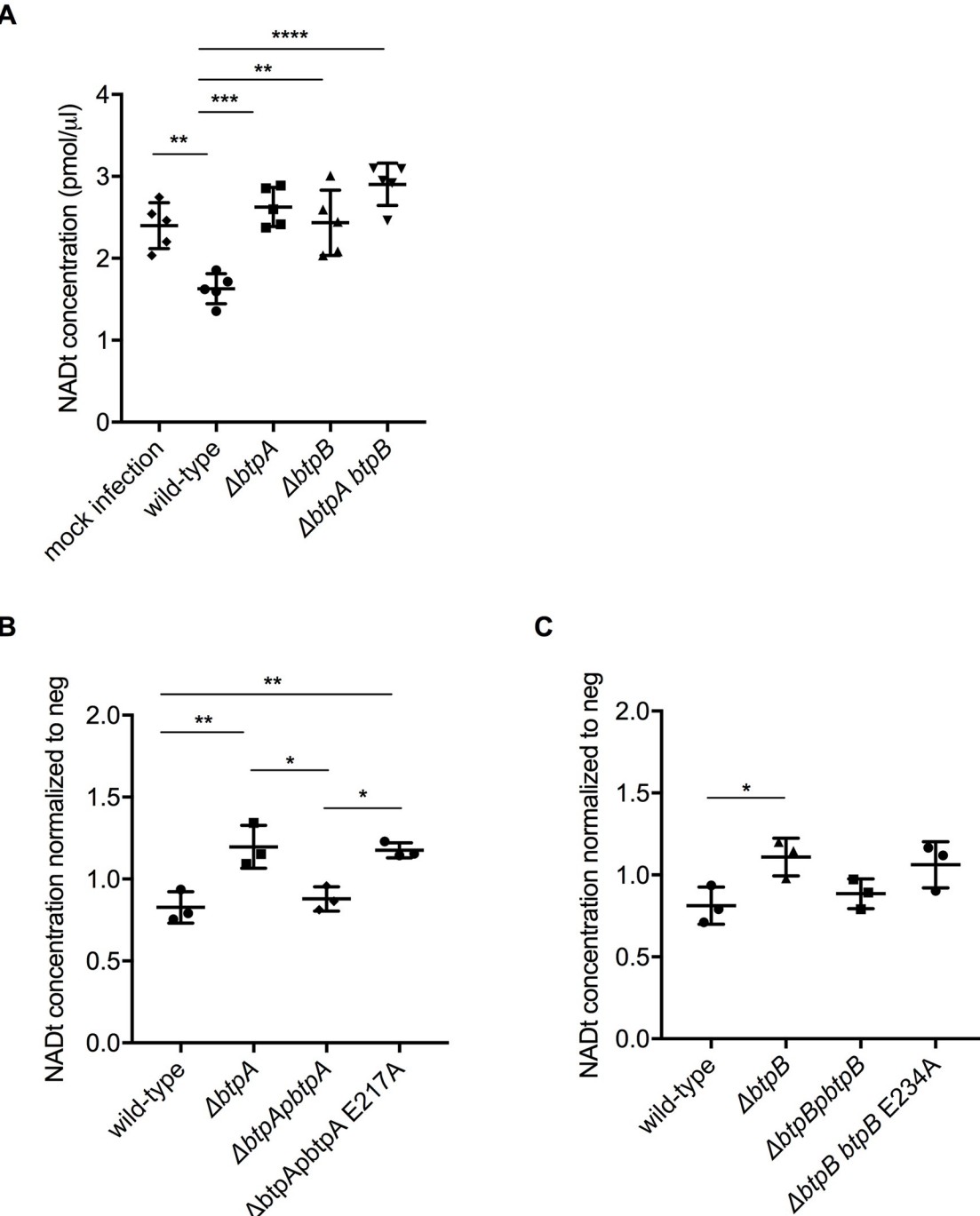

**Fig 9. *B. abortus* TIR domain-containing proteins control intracellular total NAD levels during infection of cultured epithelial cells.** (A) HeLa cells were infected for 48 h with either wild-type *B. abortus*, or strains lacking *btpA*, *btpB* or both genes. Mock infected cells are also included. Total NAD levels were measured using a colorimetric assay and data correspond to means ± standard deviation from five independent experiments and statistical comparison was done with a one-way ANOVA test, with statistical significance indicated in the graph. Between mock and wild-type p = 0.0028 (**); between the wild-type and *btpA*, *btpB* or *btpAbtpB* mutans p = 0.0002 (***), 0.0018 (**) and <0.0001 (****), respectively. Higher NAD levels in the infected cells than in the mock experiment are likely attributable to the fact that intracellular NAD levels from bacterial cells are added up to those of the cell line. (B) HeLa cells were infected for 48 h with either wild-type *B. abortus*, a Δ*btpA* mutant, the complemented strain Δ*btpA*pbtpA and a Δ*btpA* complemented with a catalytic mutant Δ*btpA*pbtpA E217A. Results are normalized to mock infected cell values and correspond to means ± standard deviation from three independent experiments and statistical comparison was done with a one-way ANOVA test, with statistical significance indicated in the graph. For wild-type *versus* Δ*btpA* p = 0.0052, Δ*btpA versus*

*ΔbtpApbtpA* 0.0124, *ΔbtpApbtpA versus ΔbtpApbtpA* E217A 0.0177 and wild-type *versus ΔbtpApbtpA* E217A 0.0072. (C) HeLa cells were infected for 48 h with either wild-type *B. abortus*, a *ΔbtpB* mutant, the complemented strain *ΔbtpBpbtpB* and a *ΔbtpB* complemented with a catalytic mutant *ΔbtpBpbtpB* E234A. Results are normalized to mock infected cell values and correspond to means ± standard deviation from three independent experiments and statistical comparison was done with a one-way ANOVA test, with a slight statistical significance only observed between wild-type and *ΔbtpB* (p = 0.0491).

We next measured the concentration of total NAD in both HeLa and iBMDM infected cells. In HeLa cells, catalytically inactive BtpA failed to complement the mutant strain (Fig 9B). Consistently, this is also the case in iBMDM for both BtpA and BtpB (Fig 10A and 10B). Together these results indicate that BtpA and BtpB contribute to depletion of intracellular total NAD levels via direct enzymatic cleavage of this metabolic co-factor during infection, assigning a novel function for these two effectors during *Brucella* infection.

## Discussion

Bacterial TIR domain-containing proteins have been shown to be major contributors to the evasion of innate immunity for a variety of bacterial pathogens, mainly by interfering with the assembly of innate immune signaling complexes involving TIR domains [4]. However, certain

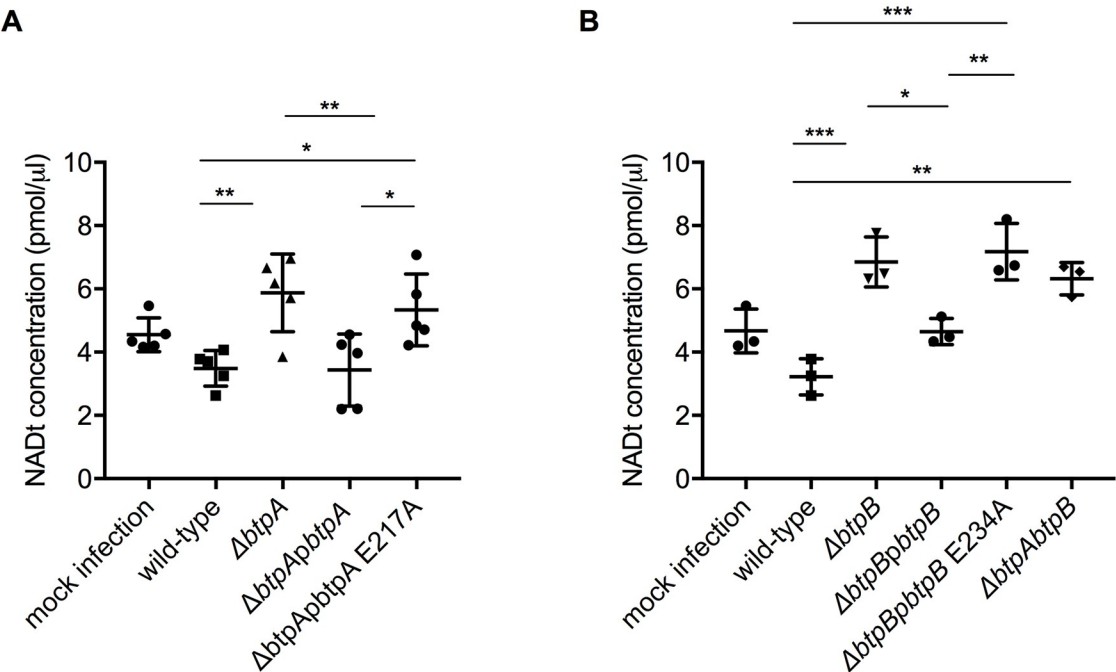

**Fig 10. *B. abortus* TIR domain-containing proteins control intracellular total NAD levels during macrophage infection.** (A) and (B) Immortalized bone marrow-derived macrophages (iBMDM) were infected for 48 h with either wild-type *B. abortus* or strains lacking *btpA*, *btpB* or both genes or complemented strains expressing *btpA* and *btpB* or the corresponding catalytic mutants. Mock infected cells are also included as a control. Total NAD levels were measured using a colorimetric assay and data correspond to means ± standard deviation from five (A) or three (B) independent experiments and statistical comparison was done with a one-way ANOVA test, with statistical significance indicated in the graph. Higher NAD levels in the infected cells than in the mock experiment are likely attributable to the fact that intracellular NAD levels from bacterial cells are added up to those of the cell line. In (A) for wild-type *versus ΔbtpA* p = 0.0071, wild-type *versus ΔbtpApbtpA* E217A p = 0.0476, *ΔbtpA versus ΔbtpApbtpA* p = 0.0058 and *ΔbtpApbtpA versus ΔbtpApbtpA* E217A p = 0.0398. In (B) for wild-type *versus ΔbtpAbtpB* p = 0.0011, wild-type *versus ΔbtpB* p = 0.0003, wild-type *versus ΔbtpBpbtpB* E217A p = 0.0001, *ΔbtpB versus ΔbtpBpbtpB* p = 0.0156 and *ΔbtpBpbtpB versus ΔbtpBpbtpB* E217A p = 0.0058. Statistical significances in relation to the negative control are not shown.

TIR domains have recently been demonstrated to possess a NAD$^+$ hydrolase activity which may contribute to their function, as for example the case of mammalian SARM1 [12] or plant NLR immune receptors [11]. We have found that both the *Brucella* TIR domain-containing proteins BtpA and BtpB retain this NAD$^+$ hydrolase activity inside cells when ectopically expressed in yeast or human cells, as well as during infection, resulting in reduction of intracellular total NAD levels at late stages of the infection. Furthermore, we have highlighted that the N-terminal non-TIR domains of these proteins are necessary for intracellular targeting of the effectors. Remarkably, BtpA-TIR and BtpB-TIR resulted in formation of long filament-like structures when ectopically expressed in yeast and human cells. Since this phenomenon can only be observed in the absence of their N-terminal regions it is likely that such N-terminal extensions may function to modulate intrinsic TIR self-assembly. Furthermore, genetic analyses in yeast revealed that such highly ordered structures formed by expression of the TIR domains alone are still achieved when expressing point mutants that lose their NADase activity, suggesting that distinct features of TIR domains rely in different structural determinants.

NAD$^+$ is an important coenzyme participating in hundreds of enzymatic reactions, notably glycolysis, the TCA cycle and mitochondrial oxidative phosphorylation. NAD$^+$ homeostasis is essential for metabolic balance and cell survival either being used as an electron carrier in redox reactions or being consumed as a substrate for numerous reactions. Beyond its well-known role in bioenergetics, NAD$^+$ has been found to have a prominent function in cell signaling, with sirtuins, poly ADP-ribose polymerases (PARPs) and CD38 using NAD$^+$ as substrate [31, 32]. NAD$^+$ has also been shown to be a key modulator of immune metabolism, acting as an important metabolic switch. In macrophages, it has been shown that increased NAD$^+$ levels are associated with activation and control of inflammatory responses, particularly involving regulation of TNFα transcription in classically activated pro-inflammatory (M1) macrophages [33, 34]. Interestingly, NAD$^+$ limitation also prompts important cellular changes such as the Warburg effect, a cellular state in which consumption of glucose is increased and aerobic glycolysis is favoured instead of the more energy efficient mitochondrial oxidative phosphorylation [35]. A switch to Warburg metabolism has also been observed upon immune activation of many cell types, for example macrophages, following pattern recognition receptor activation[36]. In addition, low NAD$^+$ levels are a trigger for cell death via necroptosis in macrophages [37].

Our work highlights that *Brucella* is decreasing total NAD levels in the host cell, likely contributing to modulation of cellular metabolism and signaling. This is dependent on two translocated effectors, BtpA and BtpB, containing a TIR domain that had previously been shown to down-modulate innate immune signaling in specific *in vitro* differentiated mouse bone marrow-derived dendritic cells [13, 14]. It is possible that the two phenotypes, NAD reduction and blocking of TIR-TIR interactions along the TLR signaling pathways are intimately connected. Indeed, targeting of these *Brucella* effectors to the vacuolar membrane or innate immune signaling platforms might locally impact NAD$^+$ levels inhibiting specific enzymatic reactions. Interestingly, the first enzyme to use NAD$^+$ in glycolysis, glyceraldehyde 3-phosphate dehydrogenase (GAPDH) has been shown to be recruited to the membrane of *Brucella*-containing vacuoles playing an essential role in intracellular replication [38]. Previous studies have also reported the role of the specific T4SS effector BPE123 in targeting the host enolase, another enzyme of the glycolysis pathway, which was also shown to be essential for *Brucella* intracellular multiplication in human cultured epithelial cells [39].

Host metabolism during *Brucella* infection has only recently started to be unravelled. In classically activated macrophages, *Brucella* infection was shown to induce a Warburg-like effect, with high consumption of glucose and generation of lactate efficiently used as a carbon source by intracellular replicating bacteria [40]. Interestingly, in alternatively activated

macrophages, abundant during chronic brucellosis, a shift from oxidative metabolism of glucose to oxidation of fatty acids occurs, enhancing the availability of glucose to promote intracellular bacterial replication [41]. In this study, we now highlight the role of the innate immune regulator effectors BtpA and BtpB in direct control of host energy metabolism.

Both BtpA and BtpB TIR domains were robust enough as NADases in the yeast heterologous model as to drop $NAD^+$ levels over one order of magnitude, causing a severe decrease of ATP availability in the cell and strong toxicity when overexpressed. The fact that full length BtpA is not as toxic for the yeast cell suggests that the N-terminal domains of BtpA negatively regulate BtpA catalytic activity. BtpB, on the contrary, is intrinsically active both in the absence and in the presence of its N-terminal extension. We show here that catalytically dead mutants in BtpA and BtpB TIR domains, still have the ability to self-aggregate and form filaments, proving that both features can be segregated. BtpA has been related to tubulin structures in host cells, specifically by protecting microtubules from depolymerization [19, 42]. However, we did not see coincidence of TIR cytoplasmic filaments with tubulin. ATP is necessary to achieve depolymerization of microtubules by nocodazole [43], so $NAD^+$ depletion and low ATP levels could contribute to the microtubule-stabilizing properties assigned to BtpA. Similar ATP-dependent phenomena could account for the inhibition of endocytosis in yeast and human cells. It is important to note however, that no impact of BtpA and BtpB on endocytosis was observed in infected cells. Therefore, ectopic over-expression of these effectors and its strong effect on ATP and NAD could be responsible for this phenotype, absent when a much smaller amount of protein is translocated into host cells during infection. Alternatively, *Brucella* infection may induce compensatory effects that would mask this phenotype, for example translocate other effectors that would enhance endocytosis.

Why *Brucella* may limit $NAD^+$ levels and energy metabolism in particular subcellular compartments and stages of the establishment of the intracellular niche? This is a challenging question and at this stage we can only speculate. NADase activity may contribute to evading the innate immune response. Importantly, $NAD^+$ levels are sensed by sirtuin proteins, like SIRT1, leading to the activation of several signaling pathways, some related to immunomodulation [44]. $NAD^+$-dependent SIRT1 has recently been shown to be an important hub for cellular defence against *M. tuberculosis* intracellular survival. *M. tuberculosis* infection reduces intracellular $NAD^+$ and down-regulates SIRT1, which can be reversed by the addition of SIRT1-activating compounds that represent a potential therapeutic option [45]. The Tuberculosis Necrotizing Toxin (TNT) of *M. tuberculosis* bears $NAD^+$ glycohydrolase activity, which was shown to be important for mycobacterial replication in macrophages, and is involved in triggering necroptosis as a consequence of $NAD^+$ depletion [37, 46]. In addition, it has been proposed that reduced $NAD^+$ and ATP levels in *Salmonella*-infected macrophages, a process dependent on the *Salmonella* Pathogenicity Island 2 (SPI-2) type 3 secretion system [47], trigger the modulation of TORC1 to protect intracellular bacteria from xenophagy via lysosomal degradation of its upstream SIRT1/LKB1/AMPK regulators [47]. Thus, $NAD^+$ depletion could eventually be understood as a strategy to evade cellular innate immunity responses and promote bacterial intracellular survival.

To our knowledge, this is the first report on bacterial TIR domain-containing proteins showing $NAD^+$ hydrolase activity on the host cell during infection. Our results on the comparison of the divergent behaviour of full length and TIR domains alone in the case of BtpA and BtpB, together with the observation that the N-terminal non-TIR extensions determine subcellular localization and prevent filament formation, suggests that these extensions may finely tune NADase activity in the context of infection, likely by negatively modulating TIR-TIR assembly and by directing NADase activity to specific intracellular compartments. Further work is now necessary to better understand the control of $NAD^+$ homeostasis during *Brucella*

infection. Knowledge on how metabolic switches occur during infection at a molecular level could provide clues for the development of therapeutic strategies and vaccines.

## Materials and methods

### *Saccharomyces cerevisiae* strains and growth conditions

YPH499 (*MATa ade 2–101 trp1-63 leu2-1 ura3-52 his3-200 lys2-801*) [48] was the *S. cerevisiae* strain for general use in these studies, unless otherwise stated. W303-1A (*MATa leu2-3,112 trp1-1 can1-100 ura3-1 ade2-1 his3-11,15*) [49] was used for the Yeast ORF overexpression library screening. The *E. coli* strain DH5α F′(K12Δ(*lacZYA-argF*)U169 *deoR supE44 thi-1 recA1 endA1 hsdR17 gyrA96 relA1* (*ϕ80lacZΔM15*)F′) was used for general molecular biology. As a general medium, YPD (1% yeast extract, 2% peptone and 2% glucose) broth or agar was used for growing yeast cells. For plasmid selection, synthetic complete medium (SC) contained 0.17% yeast nitrogen base without amino acids, 0.5% ammonium sulphate, 2% glucose and the appropriate amino acids and nucleic acid bases supplements. SG and SR were SC with 2% galactose or 1.5% raffinose, respectively, instead of glucose. For galactose induction experiments in liquid media cells were grown in SR medium to log phase and then galactose was added to 2% for 4–6 h. Effects of the expression of *Brucella* genes on yeast growth were tested by ten-fold serial dilution assay: spotting cells onto SC or SG plates lacking the appropriate auxotrophic markers to maintain the corresponding plasmids, and incubating them at 30 ºC for 72 h [26].

### Cell culture and transfections

HeLa cells (from ATCC) were grown in DMEM supplemented with 10% of fetal calf serum (FCS) and were transiently transfected using Fugene (Roche) for 24 h, according to manufacturer's instructions. Immortalized bone marrow-derived macrophages from C57BL/6J mice were grown in DMEM supplemented with 10% of fetal calf serum (FCS) and 10% of spent supernatant of L929 cells (that provides M-CSF). RAW 264.7 macrophages (ATCC) were grown in DMEM supplemented with 10% of fetal calf serum (FCS). The *Brucella abortus* 2308 strain was used for transfection and genetic manipulation.

### Construction of yeast expression plasmids

General molecular biology techniques, as well as transformation of yeast by the lithium acetate method were performed according to standard procedures. All plasmids and oligonucleotides used in this work are listed in S1 and S2 Tables respectively. pYES2-GFP, pYES3-GFP-Akt1 and YCpLG-PI3Kα-CAAX were previously described [26, 29, 50]. To clone *B. abortus btpA* into the *URA3*-based pYES2-GFP plasmid, this gene was amplified from pGEM-T-easy-BtpA plasmid [13]. The primers used for amplification of *btpA* (named BtpA-UP and BtpA-LO) had *Bam*HI respectively and *Eco*RI sites. PCR products were cleaved by these restriction enzymes to be inserted in the same sites of pYES2-GFP, generating the pYES2-GFP-BtpA plasmid. To obtain the pYES2-GFP-BtpB construction, *btpB* was amplified form pGEM-T-Easy-BtpB plasmid [13]using primers BtpB-UP and BtpB-LO, cleaved with *Bam*HI and *Xba*I restriction enzymes and the insert obtained was cloned into the pYES2-GFP plasmid. pYES3-GFP-BtpB was generated on a similar way but using primers BtpB-UP-pYES3 and BtpB-LO-pYES3 and *Bam*HI sites to clone into the *TRP1*-based pYES3-GFP plasmid. The latter plasmid had been constructed by subcloning the GFP sequence of pYES2-GFP with *Hin*dIII/*Bam*HI sites into pYES3 (Invitrogen).

In order to obtain the truncated versions of both BtpA and BtpB, we generated pYES2-GFP-BtpA-N (1–126), pYES2-GFP-BtpA-TIR (127–275), pYES2-GFP-BtpB-N (1–139) and pYES2-GFP-BtpB-TIR (140–292) by amplifying the corresponding DNA fragments from pYES2-GFP-BtpA and pYES2-GFP-BtpB using for the N-terminal regions BtpA-UP + BtpA-126stop-LO and BtpB-UP + BtpB-140stop-LO primers, respectively. In the case of the C-terminal regions, BtpA-BamHI127-UP + BtpA-LO and BtpB-BamHI140-UP and BtpB-LO primers were used. All upper primers carried *Bam*HI restriction site and lower primers had *Eco*RI site, except in the case of BtpB-LO, which carried an *Xba*I sequence. The PCR products were cleaved with their corresponding restriction enzymes and inserted in the same sites in the pYES2-GFP plasmid. Additionally, both C-terminal regions were also cloned into pYES3-GFP vector: BtpA-TIR (127–275) fragment was directly subcloned from pYES2-GFP-BtpA, whereas BtpB-TIR (140–292) was PCR amplified with BtpB-BamHI140-UP and BtpB-EcoRI-LO primers and then inserted on BamHI-EcoRI sites on pYES3-GFP.

*Dpn*I-based site directed mutagenesis was performed using the QuikChange kit (Agilent). To generate the catalytically inactive mutants in both full length and TIR domain of BtpA and BtpB in the pYES2-GFP and pYES3-GFP backbones, the mutations were introduced with primers Mut-BtpAE217A-UP and Mut-BtpAE217A-LO and Mut-BtpBE234A-UP and Mut-BtpBE234A-LO respectively. To introduce the three selected loss-of-function BtpB mutations into the TIR domain construct, primers Mut-BtpBD158G-UP andMut-BtpBD158G-LO, Mut-BtpBS162P-UP and Mut-BtpBS162P-LO, or Mut-BtpBY225C-UP and Mut-BtpBY225C-LO were used.

## Construction of mammalian expression vectors

The DNA fragment encoding amino acid residues 1–139 of BtpB (BtpB-N), 140–292 of BtpB (BtpB-TIR) and BtpB full-length from *Brucella abortus* were cloned into the Gateway pDONR (Life Technologies, ThermoFisher Scientific) and then cloned into the pENTRY (Life Technologies, ThermoFisher Scientific) GFP vectors according to the manufacturer. The following primers were used: BtpB Fw, BtpB Rv, BtpB-TIR Fw and BtpB-N Rv. BtpB E234A was constructed as above from the pYES3-GFP-BtpB E234A with primers used to amplify BtpB.

## Construction of *Brucella* complementing vectors

The DNA fragments encoding *btpA*, and *btpB* were cloned into the plasmid pBBR-1-MCS4. The primers used for amplification of *btpA* had for the forward primer a SpeI restriction site and for the reverse primer a EcoRI restriction site. The primers used for amplification of *btpB* had for the forward primer a SacI restriction site and for the reverse primer a SpeI restriction site. The following primers were used BtpA Fw-pBBR-1-MCS4, BtpA Rv-pBBR-1-MCS4, BtpB Fw-pBBR-1-MCS4, BtpB Rv-pBBR-1-MCS4. The PCR products were cleaved with their corresponding restriction enzymes and inserted in the same site in the digested pBBR-1-MCS4 plasmid. The BtpA E217A and BtpB E234A complementation vectors were obtained from pBBR-1-MCS4-BtpA and pBBR-1-MCS4-BtpB, respectively using QuickChange Site-Directed Mutagenesis.

The mutations were introduced with primers BtpA$_{E217A}$ Fw, BtpA$_{E217A}$ Rv, BtpB$_{E234A}$ Fw, BtpB$_{E234A}$ Rv.

To clone BtpA E217A and BtpB E234A into the pFlag-TEM, this genes were amplified from pBBR-1-MCS4-BtpA E217A and pBBR-1-MCS4-BtpB E234A, respectively. The primers used for amplification of *btpA$_{E217A}$* and *btpB$_{E234A}$* had for the forward primer a XbaI restriction site and for the reverse primer a PstI restriction site. The PCR products were cleaved with their corresponding restriction enzymes and inserted in the same site in the pFlag-TEM plasmid.

The primers sued were: TEM-BtpA$_{E217A}$ FW-pFlagTEM, TEM-BtpA$_{E217A}$ Rv-pFlagTEM, TEM-BtpB$_{E234A}$ FW-pFlagTEM, TEM-BtpB$_{E234A}$ Rv-pFlagTEM.

## Yeast cells microscopy and immunofluorescence

For fluorescence microscopy of live yeast to visualize GFP, cells were cultured in SR medium for 18 h at 30 ˚C, then, the appropriate amount of these cultures was suspended into fresh SG to reach an OD$_{600}$ of 0.3, and they were incubated for additional 4–6 h for *GAL1* promoter induction. Cells were harvested by centrifugation and observed directly. To monitor vacuolar morphology and endocytosis, staining with FM4-64 was performed as described [21]. Nuclear labelling was performed by adding DAPI at 1:1000 directly to the harvested cells *in vivo* and washed once with PBS. To observe actin, yeast cells were fixed and treated with rhodamine-conjugated phalloidin (Sigma) as described [23].

Indirect immunofluorescence on yeast cells was performed as previously described [51]. Antibodies were used as follows: As primary antibody, monoclonal rat anti-alpha-tubulin (Serotec, YOL1/34) at 1:500 dilution; as secondary antibody, Alexa Fluor 594 anti-rat dye (Life Technologies) at 1:1000 dilution. DAPI was added at 1:1000 for nuclear labelling. Cells were examined in Eclipse TE2000U microscope (Nikon, Tokyo, Japan) and digital images were acquired with an Orca C4742-95-12ER charge-coupled-device camera (Hamamatsu Photonics, Hamamatsu City, Japan) and processed by HCImage and ImageJ software.

## HeLa cells immunofluorescence, antibodies and microscopy

Cells were fixed in Antigenfix (DiaPath), at room temperature for 10 min or methanol at -20 ˚C for 3 min for tubulin staining. Cells were then labelled at RT with primary antibody mix diluted in 0.1% saponin in PBS with 1% BSA and 10% horse serum for blocking. Primary antibody was incubated for 1 h followed by two washes in 0.1% saponin in PBS. Secondary antibodies where then mixed and incubated for a further 30 min, followed by two washes in 0.1% saponin in PBS, one wash in PBS and one wash in distilled water before mounting with Prolong Gold. Samples were examined on a Zeiss LSM800 laser scanning confocal microscope for image acquisition. Images of 1024×1024 pixels were then assembled using ImageJ.

Primary antibodies used were mouse anti-beta-tubulin clone E7 (Developmental Studies Hybridoma Bank) at 1:250 or mouse anti-vimentin (V9) at 1:100 (Sigma). Secondary antibodies used were anti-rabbit or mouse conjugated with Alexa Fluor 488, 555 or 647 all from Jackson Immunoresearch. When necessary phallodin-568 (1:1000) was used to label the actin cytoskeleton and DAPI nuclear dye (1:1000) for the host cell nucleus. When indicated, cytochalasin D was added for 2 h at a final concentration of 1 µg/ml.

## HeLa cells endocytosis assay

Transferrin conjugated with Alexa Fluor 568 (Invitrogen) and EGF conjugated with Alexa Fluor 555 (Invitrogen) were added at 10 µg/ml and 50 µg/ml respectively, for 10 min a 37 ˚C. Cells were then placed on ice, washed with ice cold PBS twice and fixed in 3% paraformaldehyde for 15 min, followed by three washes with PBS.

## Immunodetection by western blotting

Standard procedures were used for yeast cell growth, collection, breakage, protein separation by SDS-PAGE, and transfer to nitrocellulose membranes. Anti-P-MAPK antibody (Anti-phospho-p44/ p42 MAPK (Thr-202/Tyr-204), New England Biolabs) was used to detect dually phosphorylated Slt2, Kss1 and Fus3 MAPKs diluted 1:1000. Slt2 protein was detected using a

polyclonal anti-Slt2 antibody [52], diluted 1:1000. To detect phosphorylated Hog1 high osmolarity pathway MAPK, antibody Anti-P-p38 (Sigma) at 1:1000 was used. Heterologous expressed Akt1 was detected with Anti-Akt1 (Cell Signalling) as total protein and with anti-P-Akt1(Thr)308 (Cell Signalling) for the phosphorylated forms. GFP fusion proteins were detected using monoclonal anti-GFP antibodies (Living Colors, JL-8) diluted 1:2000. As loading control either a monoclonal anti-actin (MP Biomedicals) diluted 1:2000 or a yeast specific polyclonal anti-Glucose-6-Phosphate Dehydrogenase (Sigma) diluted 1:50000 were used. In all cases, primary antibodies were detected using IRDye-680 or -800 anti-rabbit or anti-mouse antibodies (Li-Cor Biosciences), or Alexa-680 anti-mouse (Invitrogen) with an Odyssey Infrared Imaging System (Li-Cor Biosciences).

### Yeast whole genome ORF overexpression library screening

A pooled *S. cerevisiae* whole genome ORF library (Yeast ORF collection, GE Healthcare), 4500 *URA3*-based plasmids, for overexpression under *GAL1* promoter and protein A-tagged, was split into three groups. W303-1A wild type yeast strain (was co-transformed with pYES3-GFP-BtpB and one of the three library pools (S3A Fig). BtpB toxicity suppression by overexpression of a specific cDNA was tested by its ability to grow in SG agar plates. 20 different candidates were selected and tested for specificity by co-transformation with YCpLG-PI3Kα-CAAX [26], another toxic construct for yeast cells that acts by a different mechanism. Eventually 7 positive ORF, listed on S3 Table, were found to specifically rescue BtpB toxicity.

### Random mutagenesis of *btpB* and isolation of mutants

The region of pYES2-GFP-BtpB including amino acids 118 to 292 delimited by the mutazarUP and mutazarLO primers was amplified by PCR using low-fidelity Taq DNA Polymerase (Biotools) under standard conditions. The PCR products were purified with a QIAquick Gel Extraction kit (250) kit (Qiagen) and 5 μg of DNA were co-transformed into YPH499 yeast cells with 1 μg of the largest fragment of pYES2-GFP-BtpB plasmid, resulting from digestion with *Bsi*WI/*Xba*I. Such *Bsi*WI/*Xba*I digestion of pYES2-GFP-BtpB produces a gap, so the *btpB* allele can only be reconstructed upon recombination with the amplicon by *in vivo* gap repair. Recombinants were recovered by positive selection, plating the transformation mixture onto galactose-based agar medium. The pYES2-GFP-BtpB-derived plasmids were isolated from growing clones, amplified in *E. coli*, verified by restriction analysis and transformed again in yeast cells to verify that they had lost the ability to inhibit yeast cell growth. Mutations were identified on the positive clones by DNA sequencing.

### Yeast cellular ATP measurement by luciferase assay

ATP levels were measured using ENLITEN ATP Assay System (Promega) following manufacturer´s instructions. Yeast cells were cultured in SR for 18 h and then new SG was added for *GAL1*-driven gene expression to a final $OD_{600}$ of 0.3 and cultured for 3 h at 30 ˚C. Approximately $1.8 \times 10^7$ cells were harvested in 3 mL of culture and then concentrated by centrifugation for 3 min at 2500 rpm at 4 ºC. Pellets were washed with 1 mL PBS at 4 ºC and stored at -80 ºC for further analysis. For ATP extraction, pellets were resuspended with trichloroacetic acid (TCA, 5%, 10 μL) and immediately neutralized using 500 μL of Tris-acetate-EDTA buffer (TAE 1 ×; 40 mM Tris base, 20 mM acetic acid, and 1 mM EDTA, pH 7.75). The samples were centrifuged for 15 sec at 13000 rpm and then 1:100 diluted in more TAE 1×. Ten μL of this solution was mixed with 100 μL of the rL/L reagent provided by the kit and luminescence was measured using OPTOCOMP1 luminometer (MGM instruments). A standard curve for quantification was prepared using the kit´s reagents.

## Yeast cellular NAD⁺ measurement by mass spectrometry

Yeast cells were cultured as stated for the ATP luciferase assay. Approximately $6 \times 10^7$ cells were harvested in 10 mL of culture and then concentrated by centrifugation for 3 min at 2500 rpm at 4 °C. Pellets were washed with 1 mL PBS at 4 °C and stored at -80 °C for further analysis.

Our yeast NAD⁺ extraction protocol is a simplified version of the one described by Sporty et al. [53]. Pellets were resuspended in ammonium acetate (600 μL of 50 mM in MS grade water) and approximately 300 μL of 0.5–0.75 mm diameter glass beads (Reesch) were added to the tube. Cells were bead blasted at 5.5 m/s using a FastPrep-24 (MP Biomedicals) for 30 sec twice, allowing a 5 min incubation on ice in between. Supernatant was recovered by perforation of the bead-blasting tube´s base with a red-hot 0.9 x 40 mm needle. The pierced tube was placed inside a capless 1.5 mL microfuge tube and both tubes were centrifuged together for 3 min at 2000 rpm at 4 °C. This first cell lysate was stored in a new 1.5 microfuge tube on ice. The glass beads in the bead-blasting tube were then washed one more time with 600 μL of a 3:1 v/v mixture of acetonitrile (MS grade) and ammonium acetate (50 mM in MS grade water). The rinsate was then mixed with the first lysate. The mixture was clarified by centrifugation for 3 min at 13000 rpm at 4 °C and the supernatant was transferred to an ice-cold 1.5 mL microfuge tube. To standardize results, 150 μL of these lysate were kept for protein concentration measurement by Bradford method.

Samples were filtered with a 0.22 μm PTFE fliter (JASCO) and analyzed by liquid chromatography (LC) coupled to a QQQ mass spectrometer equipped with a turbo ion spray source operating in positive ion mode (LCMS 8030, Shimadzu). Chromatographic separation was performed on a Gemini C18 analytical column (50 mm×2.1 mm I.D., 2.7 μm particle size; Poroshell 120 PhenylHexyl). Injection volume was 10 μL. Samples were delivered over 11 min at a flow rate of 0.3 mL/min through the analytical column at 45 ˚C. The mobile phase was composed of A (3% methanol, 10 mM tributylamine, 3 mM acetic acid in water LC grade, 0.1% formic acid in water) and B (methanol). Mobile phase composition began with 0% B and was increased to 45% B in 2 min, to 50% in 5 more minutes and up to 95% in one minute. The mobile phase was then maintained at 95% B for 2 min and followed by re-equilibration with 0% B over the next 2 min, before injection of the next sample. Quantification of NAD⁺ was performed by multiple reactions monitoring (MRM) mode to monitor the parent ion-product ion (m/z) of the analyte. Mass transitions of m/z 662.10 to 540.00 (CE = +16 V) were used for quantification and m/z 662.10 to 407.90 (CE = +30 V) for identification with a dwell-time of 100 ms. The calibration curve was determined by plotting the peak area of the analyte (Y) versus the nominal concentration (X) with least square linear regression. All analyses were made under ISO 9001:2008 quality management system certification.

## *Brucella* infection of HeLa and iBMDM cells

Immortalized Bone Marrow-Derived Macrophages (iBMDM) were obtained as previously described [54]. HeLa cells were seeded at $1 \times 10^5$ cells/well for HeLa or $0.8 \times 10^5$ cells/well for iBMDM (6 - well plates) overnight and tryptic soy broth *Brucella* cultures inoculated and incubated for 16 h with agitation at 37 ˚C. Cells were inoculated at an MOI of 500, centrifuged at 400 g for 10 min and incubated for a further 1 h at 37 ˚C for HeLa cells and 30 min for iBMDM and RAW cells. Cells were then washed 3 times with media and incubated for 1 h with media containing gentamycin 50 μg/ml and streptomycin 100 μg/ml. After this time media was replaced to reduce the gentamycin concentration to 10 μg/ml streptomycin 20 μg/ml.

## Total NAD colorimetric assay

Total NAD (NAD+ and NADH) was extracted and quantified from cell lysates (from 2 wells of a 6-well plate for each sample) using the NAD+/NADH Colorimetric Assay Kit (Abcam, ab65348) following the manufacturer's instructions. Briefly, the amount of total NAD was calculated from a standard curve (pmol) divided by the sample volume added to the reaction well (μl) and multiplied by the dilution factor.

## TEM assay

RAW cells were seeded in a 96 well plate at $1\times10^4$ cells/well overnight. Cells were then infected with an MOI of 500 by centrifugation at 4 ˚C, 400 g for 5 min and 1 at 37 ˚C 5% $CO_2$. Cells were washed with HBSS containing 2.5 mM probenicid. Then CCF2 mix (as described in the Life Technologies protocol) and probenicid were added to each well, and incubated for 1.5 h at room temperature in the dark. Cells were finally washed with PBS, fixed using 3% PFA and analysed immediately by confocal microscopy (Zeiss LSM800).

## Statistical analysis and software

All data sets were tested for normality using Shapiro-Wilkinson test. When a normal distribution was confirmed a One-Way ANOVA test with a Tukey correction was used for statistical comparison of multiple data sets and Students t-test for two sample comparison. For data sets that did not show normality, a Kruskall-Wallis test was applied, with Dunn's correction, or Mann-Whitney U-test for two sample comparison. 3D protein images were generated using PyMOL (Schödinger), taking advantage of previously published structural data of BtpA (PDB: 4LZP) and RUN1-TIR + NADP+ (PDB: 6O0W).

## Supporting information

**S1 Fig. Expression of BtpB and BtpA versions in yeast and in inhibition of yeast endocytosis by the TIR domain of BtpB.** Western blotting of YPH499 cells extracts bearing the indicated BtpA (A) and BtpB (B) versions from pYES2-GFP plasmid derivatives, using antibodies anti-GFP (upper panels), anti-P-MAPK to show dual phosphorylation of Slt2 yeast MAPK and anti-G6PDH as loading control (lower panels). (C) Normarski and fluorescence microscopy of YPH499 cells expressing GFP-BtpB, GFP-BtpB-N and GFP-BtpB-TIR after 4h induction, stained with the endocytic marker FM4-64 for 1h. Scale bars indicate 5 μm.
(PDF)

**S2 Fig. BtpA and BtpB TIR filaments are not coincident with yeast tubulin.** Indirect immunofluorescence of YPH499 yeast cells expressing GFP-BtpA-TIR, GFP-BtpB-TIR, and their corresponding E234A mutant versions from pYES2 plasmid derivatives (green). Microtubules are stained using anti-tubulin antibody (red). Nuclei are labelled with DAPI (blue). Scale bars correspond to 5 μm.
(PDF)

**S3 Fig. Inhibitory effect of BtpB on phosphorylation of yeast signaling proteins.** (A) Western blotting from cells bearing the empty vector pYES2 (control), BtpA or BtpB from pYES2-GFP plasmid derivatives, developed with anti-P-MAPK antibody to detect dually-phosphorylated Slt2, Kss1 and Fus3 (upper panel) and anti-actin to detect actin as loading control. (B) Upper part: representative immunoblot from yeast cell lysates bearing pYES2-GFP-BtpB (+) or pYES2 (-) and upon different conditions: 30ºC (control), high temperature (39ºC), pheromone (α-factor) or Congo red, using anti-P-MAPK (upper panel), anti-

Slt2 (medium panel) and anti-actin (lower panel). Lower part: densitometric measurement of WB bands corresponding to phosphorylated MAPKs Slt2, Kss1 and Fus3. The graph displays densitometric data of phosphorylated MAPKs normalized against actin and error bars show the standard deviation from three independent experiments on different transformant clones. (C) Western blotting of cells containing the pYES2 empty vector (control) or pYES2-GFP-BtpB, developed with anti-P-p38 antibody to detect MAPK Hog1 under high osmolarity. conditions (0.6M KCl). (D) Western blotting of cells expressing heterologous Akt1 (pYES3-GFP-Akt1) with either pYES2 empty vector (control) or pYES2-GFP-BtpB, using anti-P-Akt1(Thr)308 (upper panel) and anti-Akt1 antibodies. All immunoblots were performed on protein extracts from transformants of the YPH499 yeast strain after 4 h of galactose induction.
(PDF)

**S4 Fig. Partial suppression of BtpB toxicity by overexpression of yeast genes.** (A) Ten-fold serial dilution assay of yeast cells co-expressing BtpB and each of the seven suppressor ORFs isolated from a yeast genetic screen. pYES3 and pYES2 are the corresponding empty vectors for BtpB and for the overexpressed genes, respectively. (B) Western blotting of W303-1A yeast strain co-expressing GFP-BtpB and each of the proteins encoded by the suppressor genes. Antibodies anti-GFP to detect GFP-BtpB (upper panel) and Anti-G6PDH as loading control (lower panel) were used. Anti-GFP antibody allows the detection of the indicated protein A-tagged proteins due to affinity of the tag with the Fc region of IgG-type antibodies. (C) and (D) Ten-fold serial dilution assays of yeast cells co-expressing BtpB-TIR (C) or BtpA-TIR (D) and the suppressor genes. pYES3 and pYES2 are the corresponding empty vectors for BtpB- or BtpA-TIR and for the overexpressed genes, respectively.
(PDF)

**S5 Fig. Functional analyses in yeast loss-of-function mutations in conserved residues of BtpB.** (A) Alignment of protein sequences of the TIR domains of BtpB, BtpA, human SARM1 and plant RUN1. Conserved residues relevant for this study are marked with the same color code as in Fig 4, except for for the catalytic site residues W213 and E217, that are colored in pink. (B) Structure of BtpA-TIR (left; PDB: 4LZP)) and RUN1-NADP+ complex (right; PDB: 6O0W), showing the equivalent positions of residues mutated in BtpB isolated in the yeast screen. Both structures cartoons are displayed in the same orientation. Side chain of mutated residues of BtpA relevant for this study are colored as in (A). The side chains of residues of the catalytic site of RUN1 are shown as ball-and-sticks and colored in pink and the NADP+ ligand is colored in cyan. Specific atoms are colored as follows: nitrogen in blue, oxygen in red and phosphorus in orange. (C) Phenotype of selected loss-of-function BtpB mutants. Ten-fold serial dilution growth assay of YPH499 cells transformed with pYES2 empty vector and pYES2 plasmid derivatives expressing full-length BtpB wild-type and mutants D158G, S162P and Y225C, under control (Glucose) and induction (Galactose) conditions. (D) Nomarski and fluorescence microscopy images of YPH499 cells expressing the GFP-BtpB indicated mutants, after 4h induction, stained with the endocytic marker FM4-64 for 1h. Scale bars indicate 5 μm. (E) Graph from the same experiment as in C representing the percentage of cells showing both GFP and FM4-64 vacuolar signal. Results correspond to means ± standard deviation of three independent transformants (n ≥ 100) and statistical comparison was done with one-way ANOVA with a p-value < 0.0001 (***) for all four mutants *versus* wild-type.
(PDF)

**S6 Fig. Localization and effects of GFP-BtpB versions in HeLa cells.** (A) Representative micrograph of HeLa cells expressing Myc-BtpB revealed with an anti-Myc antibody (red) and

phalloidin for labelling actin (cyan). (B) GFP-BtpB (green) can be detected at intercellular contacts (arrow and zoomed image). Cells were labeled with an anti-tubulin antibody (red). (C) HeLa cells were transfected with GFP-BtpB-TIR (green) and then labelled for tubulin (red) or (D) vimentin (red). (E) Representative image of cells labelled with anti-tubulin antibody (red) expressing aggregates of GFP-BtpB E234A (green). Scale bars correspond to 5 μm.
(TIF)

**S7 Fig. Studies on endocytic function in *Brucella*-infected cells and NAD levels of wild-type and *btp* mutant bacteria.** (A) HeLa cells or (B) iBMDM infected with either wild-type or a mutant strain lacking *btpA*, *btpB* or both genes were incubated with EGF conjugated with Alexa Fluor 555 for 10 minutes and the percentage of infected cells showing uptake of this fluorescent marker quantified by microscopy at 24h post-infection. Counts correspond to individual microscopy fields, obtained from three independent experiments. Mock infected cells are included as a control (negative). Data correspond to means ± standard deviation and statistical comparison was done with one-way ANOVA test, with no statistical significance observed. (C) Control total NAD levels from the inocula, corresponding to bacterial cultures of wild-type *B. abortus*, or strains lacking *btpA*, *btpB* or both *btpAbtpB*. Data correspond to means ± standard deviation from three independent experiments and statistical comparison was done with Kruskal-Wallis test, with no statistical significance observed. (D) RAW macrophages were infected with wild-type *B. abortus* carrying N-terminal TEM-1 fused BtpA E217A or BtpB E234A catalytic mutants for 24 h. Data represents the means ± standard errors of the percentage of cells with coumarin fluorescence from 5 independent experiments, in which at least 100 cells were analyzed per experiment and condition.
(TIF)

**S1 Table. Plasmids generated and used in this work.**
(DOCX)

**S2 Table. Oligonucleotides used in this work.**
(DOCX)

**S3 Table. Yeast genes that suppress BtpB-induced toxicity when overexpressed.**
(DOCX)

**S4 Table. BtpB loss-of-function mutants found by random mutagenesis screening on yeast.**
(DOCX)

## Acknowledgments

We wish to acknowledge the Genomics and Mass Spectrometry Units at Universidad Complutense de Madrid (UCM) for sequencing and NAD⁺ analyses, respectively. We would like to thank Thomas Henry (CIRI, Lyon) for proving us with iBMDM and Diego Comerci (CONICET, Argentina) for helping us with the complementation constructs. We also thank Lucia Sastre for technical assistance in molecular cloning and yeast growth assays and members of U3 Research Lab for help and discussion.

## Author Contributions

**Conceptualization:** Julia María Coronas-Serna, Arthur Louche, María Molina, Jean-Pierre Gorvel, Víctor J. Cid, Suzana P. Salcedo.

**Formal analysis:** Julia María Coronas-Serna, Arthur Louche, María Rodríguez-Escudero, Morgane Roussin, Laurent Terradot, María Molina, Jean-Pierre Gorvel, Víctor J. Cid, Suzana P. Salcedo.

**Funding acquisition:** María Molina, Víctor J. Cid, Suzana P. Salcedo.

**Investigation:** Julia María Coronas-Serna, Arthur Louche, María Rodríguez-Escudero, Morgane Roussin, Paul R. C. Imbert, Isabel Rodríguez-Escudero, Suzana P. Salcedo.

**Supervision:** María Molina, Víctor J. Cid, Suzana P. Salcedo.

**Writing – original draft:** Julia María Coronas-Serna, Arthur Louche, María Molina, Víctor J. Cid, Suzana P. Salcedo.

**Writing – review & editing:** Julia María Coronas-Serna, Arthur Louche, María Rodríguez-Escudero, Morgane Roussin, Paul R. C. Imbert, Isabel Rodríguez-Escudero, Laurent Terradot, María Molina, Jean-Pierre Gorvel, Víctor J. Cid, Suzana P. Salcedo.

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
