## [Decision Letter · Decision Letter 0]

3 Aug 2019

Dear Dr Salcedo,

Thank you very much for submitting your manuscript "The TIR-domain containing effectors BtpA and BtpB from Brucella abortus block energy metabolism" (PPATHOGENS-D-19-01258) for review by PLOS Pathogens. Your manuscript was fully evaluated at the editorial level and by independent peer reviewers. The reviewers appreciated the attention to an important problem, but raised some substantial concerns about the manuscript as it currently stands. These issues must be addressed before we would be willing to consider a revised version of your study. We cannot, of course, promise publication at that time.

We therefore ask you to modify the manuscript according to the review recommendations before we can consider your manuscript for acceptance. Your revisions should address the specific points made by each reviewer.

(1) A letter containing a detailed list of your responses to the review comments and a description of the changes you have made in the manuscript. Please note while forming your response, if your article is accepted, you may have the opportunity to make the peer review history publicly available. The record will include editor decision letters (with reviews) and your responses to reviewer comments. If eligible, we will contact you to opt in or out.

(2) Two versions of the manuscript: one with either highlights or tracked changes denoting where the text has been changed; the other a clean version (uploaded as the manuscript file).

Additionally, to enhance the reproducibility of your results, PLOS recommends that you deposit your laboratory protocols in protocols.io, where a protocol can be assigned its own identifier (DOI) such that it can be cited independently in the future. For instructions see http://journals.plos.org/plospathogens/s/submission-guidelines#loc-materials-and-methods

We hope to receive your revised manuscript within 60 days. If you anticipate any delay in its return, we ask that you let us know the expected resubmission date by replying to this email. Revised manuscripts received beyond 60 days may require evaluation and peer review similar to that applied to newly submitted manuscripts.

Sincerely,

Jean Celli

Guest Editor

PLOS Pathogens

Renée Tsolis

Section Editor

PLOS Pathogens

Kasturi Haldar

Editor-in-Chief

PLOS Pathogens

orcid.org/0000-0001-5065-158X

Grant McFadden

Editor-in-Chief

PLOS Pathogens

orcid.org/0000-0002-2556-3526

Dear Dr Salcedo,

We are returning your manuscript with three reviews. As you will see, the reviewers all appreciated the amount and quality of the work described in the manuscript, but came to different conclusions about the relevance and significance of the paper, ranging from a “…huge and original work of high interest…” to “…potentially non-specific effects observed with overexpression of the BtpB or the TIR domains of BtpA and/or BtpB…”. We tend to agree that the study is heavily weighted towards ectopic expression approaches and requires additional experimentation using infection models to strengthen its conclusions and relevance.

After reading the reviews and looking at the manuscript, we have rendered a decision Major Revision based on the critiques from Reviewers 2 and 3. A significant concern shared by these reviewers that requires your outmost attention is the need to further corroborate the authors’ observations made using ectopic expression approaches in yeast and HeLa cells with experimental data generated using infection models, in order to establish the relevance of the BtpA and BtpB-induced phenotypes. In particular, it appears essential to:

1. confirm that the effect of BtpA and BtpB on NAD+ levels during infection depends on their TIR domain NAD hydrolase activity via genetic complementation approaches;

2. address the concern raised by Reviewer 3 about the redundant role(s) of BtpA and BtpB on NAD+ levels during infection;

3. verify that BtpA and BtpB’s effects on metabolism (and probably other phenotypes revealed in yeast) during infection is reproducible in macrophages.

We would also like to encourage you to address all the other reviewers’ comments in your revised manuscript and responses. We are sorry We cannot be more positive at the moment, however we are looking forward to receiving your revision. Note that we may send your paper back to some of the reviewers upon resubmission.

Reviewer's Responses to Questions

**Part I - Summary**

Reviewer #1: The authors report a new characterization of two Brucella effectors, BtpA and BtpB. They start by showing that these two proteins are toxic for yeast, which depends on their TIR domain. They also report that BtpB generate several defects in yeast, such as a blockage of endocytosis. After screening for suppressors, they realize that NAD+ and ATP levels are affected in yeast producing BtpA or BtpB. They used elegant controls to show that it is indeed the catalytic activity of BtpB, able to degrade NAD+, that is responsible for toxicity in yeast. The authors also show that BtpB is inhibiting endocytosis in humans cells, that BtpA and BtpB can also generate a drop in the level of NAD+ in humans cells, and -importantly- they show that Brucella infection generates a decrease in the NAD+ pool that is not observed in a btpA or a btpB mutant.

This is a huge and an original work of high interest, an interest that is not limited to the Brucella research community. The approach is very interesting, starting from yeast to generate focused and testable hypotheses on human cells. It also generates a new possible function for this family of effectors, suggesting that the pioneering study of Essuma et al. (2017 and 2018) is indeed applicable to several pathogens. This works thus opens new perspectives to the field of host-pathogen interactions, at the molecular and cellular levels.

I have a list of minor issues that should be taken into account for the submission of revised version of this manuscript.

Reviewer #2: This is an interesting paper that suggests the Brucella modulates host metabolic pathways through the activities of two type IV secreted effectors, BtpA and BtpB. The majority of these conclusions are based on potentially non-specific effects observed with overexpression of the BtpB or the TIR domains of BtpA and/or BtpB in yeast and mammalian cells. These phenotypes can be interpreted to be due to the recently identified NAD+ consuming activity of TIR domains- which are found in these effectors. However, it is unclear how the heterologous expression studies would have led the investigators to determine this likely function of the effectors, in the absence of the prior published studies which established BtpA as a NAD+-consuming protein. The bulk of the data presented is focused on characterizing the phenotypes observed with the overexpression of these effectors in these cells. Very little is done to actually determine their roles during an infection, including evidence that BtpA and/or BtpB directly regulate energy metabolism during an infection. In addition, no clear connection between the role of the proposed NAD-consuming activity of the TIR domains and their previously identified roles in regulating innate immune responses.

Reviewer #3: Recently Essuman et al reported that bacterial TIR domains were NAD hydrolases. In this work, the BtpA and BtpB proteins were evaluated ectopically in yeast and HeLa cells, and during infection of HeLa cells. In yeast, critical TIR residues were mapped. In yeast, the TIR domains disrupted the actin skeleton, endocytosis and multiple MAPkinase signaling events, significantly decreasing NAD+ and ATP levels. In HeLa cells, the ectopic TIR domains decreased NAD and endocytosis and protected the actin cytoskeleton from drugs. During infection, NAD notably decreased but endocytosis was not affected. These data were accompanied by confocal micrographs detailing localization of the different constructs tested. Mutants segregated capacity to form filaments and toxicity.

Overall, the work was thorough, well performed and described, and has some intriguing implications for previously undescribed effects of the Btp proteins during infection. Altered metabolism during infection is only recently been appreciated and is an intriguing topic. However, there are some concerns regarding some of the data and conclusions from this data:

1. During infection, why do the BtpA and BtpB mutations exert the same effect, completely recovering NAD+? If both are expressed during infection, shouldn’t there be only partial recovery or an additive effect in the double mutant?

2. In the discussion, the authors surmise that the lack of effect on endocytosis is cell type specific and further that phagocytes may respond differently. Since both ectopic expression (where endocytosis blocked) and infections were both performed in HeLa cells, I’m not sure I understand this argument. It would seem that the difference in endocytosis effect most likely reflects magnitude of expression. Further, it should be straightforward to infect macrophages with these mutants and determine effects on endocytosis and NAD+ levels.

3. Timing is a question: what is the time course for NAD+ suppression? Does it take until 48h to see an effect? The Btp proteins are expressed early following infection – could this be because of intracellular Brucella numbers needed to detect an effect?

4. Although not necessary, is it possible to detect an effect on glycolysis vs oxidative phosphorylation in mutant infected cells? It would be difficult to tease apart effects on cytokine production because of the other proposed TLR-antagonizing functions of the Btp TIR domains.

5. Multiple TIR domains have been crystallized. Besides being conserved, what else to these critical residues do? Are they important for folding or TIR-TIR interaction?

6. In the yeast, the whole BtpA had no effect on ATP and minimal effect on NAD+, yet during infection, increase in NAD was comparable between BtpA and BtpB mutants. Any thoughts as to why this might be the case?

**Part II – Major Issues: Key Experiments Required for Acceptance**

Reviewer #1: None

Reviewer #2: 1. It is appreciated that a significant amount of work went into producing all of the yeast data (Figures 1-6), but in retrospect what do the investigators feel that the readers can take away from all of the yeast data that has been presented? It appears that most of the observed phenotypes reflect non-specific effects due decrease ATP/NAD levels, presumably a defect that will arise with the expression of many toxic proteins. It does not appear that they would ever have come to the conclusion that BtpA and BtpB might have NAD consuming activity, if not for the prior publication in Current Biology in 2018.

2. The significance of the yeast overexpressor screen is unclear. Again, it is very unclear how the few hits found in addition to the phenotypes obsereved in yeast would have led the investigators to the discovery that BtpB is a NAD-consuming protein. The hits in this genome-wide screen are quite disparate and are only weak suppressors. It is reassuring that they don’t suppress the toxicity observed with overexpression of P13kalpha-CAAX, but this isn’t the most convincing argument that their mechanism of suppression is specific to BtpB. Do they also suppress the toxicity associated with expression of the BtpB and BtpA TIR domains? Is it surprising that they did not find overexpressor of proteins involved NAD metabolism that suppress toxicity?

3. Lines 344-350. This section is confusing. It appears that there is prior evidence that TIR expression stabilizes microtubules, is this also seen here? Or, are the investigators observing that TIR is stabilizing only actin cytoskeleton? It seems like it is the latter. Did the investigators also look at microtubule stability? This experiment is not necessary, particularly as they present no evidence for a role in BtpA or BtpB in altering the status of the cytoskeleton in the context of an infection.

4. In the infection assays, what is the rationale for assaying for changes in NAD levels at the 16 hour time point shown in Fig. 11B? At what time are BtpA and BtpB expressed in intracellular bacteria? When are they observed in the cytosol of host cells? How stable are they? The big question here is whether or not the investigators are studying a direct or indirect effect? Can the investigators complement the btpA and btpB knockouts? Do variants predicted to be catalytically dead no longer complement? The latter are important controls that should be included. Also, it doesn’t appear that the actions of BtpA and BtpB are functionally redundant. How do the investigators explain this?

5. In Fig. S6B, are the investigators growing the Brucella under conditions that would induce expression of BtpA and BtpB? If not, the significance of this control is unclear, as presumably the bacteria grow differently in vitro than within the c host cells. Can the investigators, isolate the intracellular Brucella and assess NAD levels? This would be the appropriate control.

Reviewer #3: There are no major experimental issues, though performing the experiments requested above in macrophages would significantly enhance impact, particularly as cell type is posited as a reason for differences.

Clarification of the timing of NAD+ suppression during infection would be appreciated.

Beyond these experiments, the current data requires more explanation.

**Part III – Minor Issues: Editorial and Data Presentation Modifications**

Reviewer #1: I think that the title could be more specific than "block energy metabolism", maybe mention NAD+?

lines 71 and 72 : can we really propose that bacteria have strategies and tactics? I know it is widespread, but I would avoid it

line 133 : do the authors really show that there is a blockage of the bioenergetics patways? I agree that a depletion of the pool of NAD+ will have consquences on host metabolism

line 171 : these results do not suggest that TIR are prone SELF-interact, they could interact with a filamentous structure in yeast cells

lines 2017-2018 : the western blots with-Kss1 and P-Fus3 are not really convincing in Figures S1A and S1B, are they really necessary?

line 268 : "In agreement", with what?

Fig. S4A and Table S4 : what is the mutation Y187X? did you not have frameshift mutations? or mutations to stop codons? Last column in Table S4 can be deleted (simply state somewhere that they were all isolated once).

line 301 : "tolerated" is an interpretation, be more cautious (although I agree that it is likely)

line 313 : as well as Myc-expressng BtpB?

line 315 : "A proportion", can it be quantified?

line 376 : I would suggest to move Fig. 11A to the supplementary data

line 387 : add a reference to show that "it is well established that btp mutants replicate to the same levels". Is it also true for the double deletion strain?

line 456 : the NAD+ level drops over one order of magnitude, but is it due to the overexpression of the effector?

line 479 : shown to be an important hub?

line 1049 (legend of Fig. 9) : how do you decide that a cell is "showing uptake", is it the case if only one vacuole is positive? or where do you place your cutoff to classify your cells as positive or negative for endocytosis? Please clarify

Legends of figures 10 and 11 : could you clarify the units used for the evaluation of the NAD+ level? (microliters refer to what?)

Reviewer #2: (No Response)

Reviewer #3: 1. In Fig S1, there are a lot of bands in the gel – how were the relevant ones be identified? Why are the proteins doublets? In this Figure, the Slt2 band is cropped so tightly, it is difficult to see. For the other MAPkinases, the gel has so much spotting, it is difficult to make out bands at all, and a decrease in phosphorylation vs vector control cannot be appreciated. Are there any other blots that could be substituted?

2. I could not appreciate the “much larger” E234A aggregates from Figure 8. In S5E, the cell nucleus looks much larger, even though the magnification bar is smaller – there seem to be some other discrepancies between cell size and apparent magnification – is the lowest row of Fig8 at a higher magnification?

3. Is the amount of NAD+ significantly increased in the double mutant in Fig. 11? It’s not shown.

4. Please provide the references stating that the Btp mutants replicate to the same amount.

5. For the western blots in Figure 2, could you provide number of times this experiment was done and combine to generate error bars for 2D?

6. In Figure 3, what are the “Vectors” vs “vector” row?

7. In Figure 5, it is not clear what the “double mutants” or open red triangles refer to.

PLOS authors have the option to publish the peer review history of their article (what does this mean?). If published, this will include your full peer review and any attached files.

Reviewer #1: No

Reviewer #2: No

Reviewer #3: No

---

## [Decision Letter · Decision Letter 1]

3 Mar 2020

Dear Dr Salcedo,

Thank you very much for submitting your manuscript "The TIR-domain containing effectors BtpA and BtpB from Brucella abortus impact NAD metabolism" for consideration at PLOS Pathogens. As with all papers reviewed by the journal, your manuscript was reviewed by members of the editorial board and by several independent reviewers. The reviewers appreciated the attention to an important topic.

As you will see from the attached comments, all reviewers appreciated the amount of efforts to address their original concerns to their satisfaction and do not have any major remaining issues with the revised manuscript, which they now deem an important contribution to the field. Reviewer 2 and 3 have minor suggestions and comments that all need to be addressed, however. In particular, I concur with the suggestion by Reviewer 2 to reorganize Figures 5 to 10, in order to decrease emphasis on phenotypes that are not relevant to infection (Fig. 7) and merge related data into single figures (Fig. 8-9-10).

Based on the reviews, we are likely to accept this manuscript for publication, providing that you modify the manuscript according to the review recommendations.

Sincerely,

Jean Celli

Guest Editor

PLOS Pathogens

Renée Tsolis

Section Editor

PLOS Pathogens

Kasturi Haldar

Editor-in-Chief

PLOS Pathogens

orcid.org/0000-0001-5065-158X

Michael Malim

Editor-in-Chief

PLOS Pathogens

orcid.org/0000-0002-7699-2064

Reviewer Comments (if any, and for reference):

Reviewer's Responses to Questions

**Part I - Summary**

Reviewer #1: This study generates a possible function for this family of effectors, suggesting that the pioneering study of Essuma et al. (2017 and 2018) is indeed applicable to several pathogens. This works thus opens new perspectives to the field of host-pathogen interactions, at the molecular and cellular levels.

Reviewer #2: The investigators have addressed all of my concerns adequately. The new experiments demonstrating the NAD concentration within cells infected with wt vs mutant bacteria are a great addition. My only suggestion would to tighten up the figures. These changes are not essential, but I think that doing so would place more emphasis on the significant findings. Specifically, given that figure 7 basically demonstrates likely artifactual phenotypes associated with effector overexpression, it can be moved to supplement. Similarly, figures 5 and 6 could be combined as they are both looking at localization patterns of ectopically expressed full-length and truncated effectors. Lastly fig. 8A could be move to supplement and Figures 8B, 9 and 10 combined to one figure.

Reviewer #3: In this study the authors characterize a previously undescribed function for Brucella TIR-containing proteins BtpB and BtpA in hydrolyzing NAD+. The manuscript represents an immense amount of work, and characterizes critical domains and amino acid residues for the NAD activity in yeast, HeLa cells and macrophages. The results using Brucella mutants in infected cells help validate the yeast overexpression studies. The efforts at complementation and localization of critical residues on the crystal structure add to the impact. The potential consequences of NAD+ decreases for macrophage function are intriguing. These results also have implications for other pathogens expressing TIR-containing proteins. Minor concerns are below.

**Part II – Major Issues: Key Experiments Required for Acceptance**

Reviewer #1: None, I appreciated that complementation of the mutants was performed in the revision since it was indeed an important control.

Reviewer #2: (No Response)

Reviewer #3: none

**Part III – Minor Issues: Editorial and Data Presentation Modifications**

Reviewer #1: The minor issues that I raised were managed by the authors during revision. I realized that the authors should indicate which strain of B. abortus was used in this study, unless I have missed it in the revised manuscript.

Reviewer #2: (No Response)

Reviewer #3: 1. Results question: Sometimes, complementation studies can raise more questions than they solve. In Figure 10, I do not see stats bars between mock infected and wild type - are the wild type NAD levels truly lower than mock? Also, why do deltaBtpB and possibly deltaBtpA have higher NAD+ levels than mock infected? Does this suggest that in the absence of Btp proteins, Brucella might increase NAD, which could limit infection? The reason may not be known, but these results should at least be acknowledged.

2. Method of immortalizing the macrophages should be described or referenced.

PLOS authors have the option to publish the peer review history of their article (what does this mean?). If published, this will include your full peer review and any attached files.

Reviewer #1: No

Reviewer #2: No

Reviewer #3: Yes: Judith A Smith
---

## [Editor Report · Decision Letter 2]

26 Mar 2020

Dear Dr Salcedo,

We are pleased to inform you that your manuscript 'The TIR-domain containing effectors BtpA and BtpB from Brucella abortus impact NAD metabolism' has been provisionally accepted for publication in PLOS Pathogens.

Best regards,

Jean Celli

Guest Editor

PLOS Pathogens

Renée Tsolis

Section Editor

PLOS Pathogens

Kasturi Haldar

Editor-in-Chief

PLOS Pathogens

orcid.org/0000-0001-5065-158X

Michael Malim

Editor-in-Chief

PLOS Pathogens

orcid.org/0000-0002-7699-2064
---

## [Editor Report · Acceptance letter]

10 Apr 2020

Dear Dr Salcedo,

We are delighted to inform you that your manuscript, "The TIR-domain containing effectors BtpA and BtpB from *Brucella abortus* impact NAD metabolism," has been formally accepted for publication in PLOS Pathogens.

Best regards,

Kasturi Haldar

Editor-in-Chief

PLOS Pathogens

orcid.org/0000-0001-5065-158X

Michael Malim

Editor-in-Chief

PLOS Pathogens

orcid.org/0000-0002-7699-2064